# SYNERGY BETWEEN SATELLITE OBSERVATIONS OF SOIL MOISTURE AND WATER STORAGE ANOMALIES FOR RUNOFF ESTIMATION

Stefania Camici [(1)], Gabriele Giuliani [(1)], Luca Brocca [(1)], Christian Massari [(1)], Angelica Tarpanelli [(1)], Hassan Hashemi Farahani [(2)], Nico Sneeuw [(2)], Marco Restano [(3)], Jérôme Benveniste [(4)]

*(1) National Research Council, Research Institute for Geo-Hydrological Protection, Perugia, Italy (s.camici@irpi.cnr.it)*

*(2) Institute of Geodesy, University of Stuttgart, Geschwister-Scholl-Straße 24D, 70174 Stuttgart, Germany*

*(3) SERCO c/o ESA-ESRIN, Largo Galileo Galilei, Frascati, 00044, Italy*

*(4) European Space Agency, ESA-ESRIN, Largo Galileo Galilei, Frascati, 00044, Italy*

**November 2020**

Submitted to:

\*      Correspondence to: Ph.D. Stefania Camici, Research Institute for Geo-Hydrological Protection, National Research Council, Via della Madonna Alta 126, 06128 Perugia, Italy. Tel: +39 0755014419 Fax: +39 0755014420      E-mail: stefania.camici@irpi.cnr.it.

**ABSTRACT**

This paper presents an innovative approach, STREAM - SaTellite based Runoff Evaluation And Mapping - to derive daily river discharge and runoff estimates from satellite soil moisture, precipitation and total water storage anomalies observations. Within a very simple model structure, precipitation and soil moisture data are used to estimate the *quick-flow* river discharge component while the total water storage anomalies are used for obtaining its complementary part, i.e., the *slow-flow* river discharge component. The two are then summed up to obtain river discharge estimates. The method is tested over the Mississippi river basin for the period 2003-2016 by using Tropical Rainfall Measuring Mission (TRMM) Multi-satellite Precipitation Analysis (TMPA) precipitation data, European Space Agency Climate Change Initiative (ESA CCI) soil moisture data and Gravity Recovery and Climate Experiment (GRACE) total water storage data. Despite the model simplicity, relatively high-performance scores are obtained in river discharge estimates, with a Kling-Gupta efficiency index greater than 0.64 both at the basin outlet and over several inner stations used for model calibration highlighting the high information content of satellite observations on surface processes. Potentially useful for multiple operational and scientific applications, from flood warning systems to the understanding of water cycle, the added-value of the STREAM approach is twofold: 1) a simple modelling framework, potentially suitable for global runoff monitoring, at daily time scale when forced with satellite observations only, 2) increased knowledge on the natural processes, human activities and on their interactions on the land.

Key words: satellite products, soil moisture, water storage variations, conceptual hydrological modelling, rainfall-runoff modelling, Mississippi.

## 1. INTRODUCTION

Spatial and temporal continuous river discharge monitoring is paramount for improving the understanding of the hydrological cycle, for planning human activities related to water use as well as to prevent or mitigate the losses due to extreme flood events. To accomplish these tasks, runoff and river discharge data, which represents the aggregated signal of runoff (Fekete et al., 2012), should be available at adequate spatial and temporal resolution. For water resources management and drought monitoring monthly time series over basin area larger than 10'000 $km^2$ are sufficient whereas observations up to grid scale of few km and daily or sub-daily time step are required for flood prediction. The accurate spatio-temporally continuous runoff and river discharge estimation at finer spatial or temporal resolution is still a big challenge for hydrologists.

Traditional in situ observations of river discharge, even if generally characterized by high temporal resolution (up to sub-hourly time step), typically offer little information on the spatial distribution of runoff within a watershed. Moreover, river discharge observation networks suffer from many limitations such as low station density and often incomplete temporal coverage, substantial delay in data access and large decline in monitoring capacity (Vörösmarty et al., 2002). Paradoxically, this latter issue is exacerbated in developing nations (Crochemore et al., 2020), where the knowledge of the terrestrial water dynamics deserves greater attention due to huge damages to settlements and especially the loss of human lives that occurs regularly.

This precarious situation has led to growing interest in finding alternative solutions, i.e., model-based or observation-based approaches, for runoff and river discharge monitoring. Model-based approaches, based on the mathematical description of the main hydrological processes (e.g., water balance models, WBMs, global hydrological models, GHMs, e.g., Döll et al., 2003 or, increasing in complexity, land surface models, LSM, e.g., Balsamo et al., 2009; Schellekens et al., 2017), are able to provide comprehensive information on a large number of relevant variables of the hydrological cycle including runoff and river discharge at very high temporal and spatial resolution (up to hourly

sampling and 0.05° grid scale). However, the values of modelled water balance components rely on
a massive parameterization of the soil, vegetation and land parameters, which is not always realistic,
and are strongly dependent on the GHM or LSM models used, analysis periods (Wisser et al., 2010)
and climate forcings selected (e.g Haddeland et al., 2012; Gudmundsson et al., 2012a, b; Prudhomme
et al., 2014; Müller Schmied et al., 2016).
Alternatively, the observation-based approaches exploit machine learning techniques and a
considerable amount of data to describe the physics of the system (Solomatine and Ostfeld, 2008)
with only a limited number of assumptions. Besides being simpler than model-based approaches,
these approaches still present some limitations. For example, they rely on a considerable amount of
data describing the modelled system's physics and the spatial/temporal extent and the uncertainty of
the resulting dataset is determined by both the spatial and temporal coverage and the accuracy of the
forcing data (e.g., see E-RUN dataset, Gudmundsson and Seneviratne, 2016; GRUN dataset, Ghiggi
et al., 2019; FLO1K dataset, Barbarossa et al., 2018). Additional limitations stem from the employed
method to estimate runoff. Indeed, random forests such as employed in Gudmundsson and
Seneviratne (2016) like other machine learning techniques, are powerful tools for data driven
modeling, but they are prone to overfitting, implying that noise in the data can obscure possible
signals (Hastie et al., 2009). Moreover, the influence of land parameters on continental-scale runoff
dynamics is not considered as the underlying hypothesis is that the hydrological response of a basin
exclusively depends on present and past atmospheric forcing. It is easy to understand that this
assumption will only be valid in certain circumstances and might lead to problems, e.g., over complex
terrain (Orth and Seneviratne, 2015) or in cases of human river flow regulation (Ghiggi et al., 2019).
Remote sensing can provide estimates of nearly all the climate variables of the global hydrological
cycle including soil moisture (e.g., Wagner et al., 2007; Seneviratne et al., 2010), precipitation
(Huffman et al., 2014) and total terrestrial water storage (e.g., Houborg et al., 2012; Landerer and
Swenson, 2012; Famiglietti and Rodell, 2013). It has undeniably changed and improved dramatically
the ability to monitor the global water cycle and, hence, runoff. By taking advantage of satellite
information, some studies tried to develop methodologies able to optimally produce multivariable
datasets from the fusion of in situ and satellite-based observations (e.g., Rodell et al., 2015; Zhang et
al., 2018; Pellet et al., 2019). Other studies exploited satellite observations of hydrological variables,
e.g., precipitation (Hong et al, 2007), soil moisture (Massari et al., 2014), and geodetic variables (e.g.,
Sneeuw et al., 2014; Tourian et al., 2018) to monitor single components of the water cycle in an
independent way.
Although the majority of these studies provide runoff and river discharge data at basin scale and
monthly time step, they deserve to be recalled here as important for the purpose of the present study.
In particular, Hong et al. (2007) presented a first attempt to obtain an approximate but quasi-global
annual streamflow dataset by incorporating satellite precipitation data in a relatively simple rainfall-
runoff simulation approach. Driven by the multiyear (1998-2006) Tropical Rainfall Measuring
Mission Multi-satellite Precipitation Analysis, runoff was independently computed for each global
land surface grid cell through the Natural Resources Conservation Service (NRCS) runoff curve
number (CN) method (NRCS, 1986) and subsequently routed to the watershed outlet to predict
streamflow. The results, compared to the in situ observed river discharge data, demonstrated the
potential of using satellite precipitation data for diagnosing river discharge values both at global scale
and for medium to large river basins. If, on the one hand, the work of Hong et al. (2007) can be
considered as a pioneer study, on the other hand it presents a serious drawback within the NRCS-CN
method that lacks a realistic definition of the soil moisture conditions of the catchment before flood
events. This aspect is not negligible as it is well established that soil moisture is paramount in the
partitioning of precipitation into surface runoff and infiltration inside a catchment (Brocca et al.,
2008). In particular, for the same rainfall amount but different values of initial soil moisture
conditions, different flooding effects can occur (see e.g. Crow et al., 2005; Brocca et al., 2008; Berthet
et al., 2009; Merz and Bloschl, 2009; Tramblay et al., 2010). On this line following Brocca et al.
(2009), Massari et al. (2016) presented a very first attempt to estimate global streamflow data by
using satellite Soil Moisture Active and Passive (SMAP, Entekhabi et al., 2010) and Global

Precipitation Measurement (GPM, Huffman et al., 2019) products. Although the validation was carried out by routing the monthly surface runoff only in a single basin in Central Italy, the obtained results suggested to dedicate additional efforts in this direction.

Among the studies that use satellite observations of hydrological variables for runoff estimation, the hydro-geodetic approaches are undoubtedly worth mentioning, see e.g., Sneeuw et al. (2014) for a comprehensive overview or Lorenz et al. (2014) for an analysis of satellite-based water balance misclosures with discharge as closure term. In particular, the satellite mission Gravity Recovery And Climate Experiment (GRACE), which observed the temporal changes in the gravity field, has given a strong impetus to satellite-driven hydrology research (Tapley et al., 2019). Since temporal gravity field variations over the continents imply water storage change, GRACE was the first remote sensing system to provide observational access to deeper groundwater storage. GRACE and its successor mission GRACE-FO provide monthly snapshots of the Earth's gravity field. The temporal variation is therefore relative to the temporally mean gravity field and, hence, the time variations of water storage are fundamentally relative to the mean storage. This relative water storage variation is termed Total Water Storage Anomaly (TSWA).

The relation between GRACE-derived TWSA and runoff was characterized by Riegger and Tourian (2014), which even allowed the quantification of absolute drainable water storage over the Amazon (Tourian et al., 2018). In essence, the storage-runoff relation describes the gravity-driven drainage of a basin and, hence, the slow-flow processes. Due to GRACE's spatial-temporal resolution, runoff and river discharge are generally available for large basins (>160'000 km$^2$) and at monthly time step.

Based on the above discussion, it is clear that each approach presents strengths and limitations that enable or hamper the runoff and river discharge monitoring at finer spatial and temporal resolutions. In this context, this study presents an attempt to find an alternative method to derive daily river discharge and runoff estimates at 0.25° degree spatial resolution exploiting satellite observations and the knowledge of the key mechanisms and processes that act in the formation of runoff, i.e., the role of soil moisture in determining the response of a catchment to precipitation. For that, soil moisture,

precipitation and TWSA observations are used as input into a simple modelling framework named
STREAM v1.3 (SaTellite based Runoff Evaluation And Mapping, version 1.3, hereafter referred to
as STREAM). Unlike classical LSMs, STREAM exploits the knowledge of the system states (i.e.,
soil moisture and TWSA) to derive river discharge and runoff, and thus it 1) skips the modelling of
the evapotranspiration fluxes which are known to be a non-negligible source of uncertainty (Long et
al. 2014), 2) limits the uncertainty associated with the over-parameterization of soil and land
parameters and 3) implicitly takes into account processes, mainly human-driven (e.g., irrigation,
change in the land use), that might have a large impact on the hydrological cycle and hence on runoff.
The detailed description of the STREAM model is given in section 4. The collected datasets and the
experimental design for the Mississippi River Basin (section 2) are described in section 3 and 5,
respectively. Results, discussion and conclusions are drawn in section 6, 7 and 8, respectively.
**2.  STUDY AREA**
The STREAM model presented here has been tested and validated over the Mississippi River basin
(Figure 1a). With a drainage area of about 3.3 million km$^2$, the Mississippi River basin is the fourth
largest watershed in the world, bordered to the West by the crest of the Rocky Mountains and to the
East by the crest of the Appalachian Mountains. According to the Köppen climate classification, the
climate is subtropical humid over the southern part of the basin, continental humid with hot summer
over the central part, continental humid with warm summer over the eastern and northern parts,
whereas a semiarid cold climate affects the western part. The average annual air temperature across
the watershed ranges from 4°C in the West to 6°C in the East. On average, the watershed receives
about 900 mm/year of precipitation (77% as rainfall and 23% as snowfall), more concentrated in the
eastern and southern portions of the basin with respect to its northern and western part (Vose et al.,

168    2014).

The river flow has a clear natural seasonality mainly controlled by spring snowmelt (coming from
the Missouri and the Upper Mississippi, the western and the north-central part of the basin,
respectively, Dyer 2008) and by heavy precipitation exceeding the soil moisture storage capacity
(mostly occurring in the eastern and southern part of the basin, Berghuijs et al., 2016). The basin is
also heavily regulated by the presence of large dams (Global Reservoir and Dam Database GRanD,
Lehner et al., 2011) most of them located on the Missouri river and over the Great Plains. In particular,
the river reach between Garrison and Gavins Point dams is the portion of the Missouri river where
the large main-channel dams have the greatest impact on river discharge providing a substantial
reduction in the annual peak floods, an increase on low flows and a reduction on the overall variability
of intra-annual discharges (Alexander et al., 2012). The annual average of Mississippi river discharge
at Vicksburg, the outlet river cross-section of the basin, is equal to 17'500 $m^3$/s (see Table 1). Given
the variety of climate and topography across the Mississippi River basin, it is a good candidate to test
the suitability of the STREAM model for river discharge and runoff simulation.
**3. DATASETS**
The datasets used in this study include in situ observations, satellite products and runoff verification
data. The first two datasets are used as input data to the STREAM model. Conversely, the runoff
verification data are used as a benchmark to validate the performance of the STREAM model in
simulating the runoff.
**3.1 In situ Observations**
In situ observations comprise air temperature and river discharge data.
For air temperature data the Climate Prediction Center (CPC) Global Temperature data developed by
the American National Oceanic and Atmospheric Administration (NOAA) using the optimal
interpolation of quality-controlled gauge records of the Global Telecommunication System (GTS)
network (Fan et al., 2008) have been used. The dataset is available on a global regular 0.5°×0.5° grid
and provides daily maximum ($T_{max}$) and minimum ($T_{min}$) air temperature data from 1979 to present
(2022). The daily average air temperature data have been generated as the mean of $T_{max}$ and $T_{min}$ of
each day.

Daily river discharge data over the study basin have been taken from the Global Runoff Data Center (GRDC, https://www.bafg.de/GRDC/EN/Home/homepage_node.html). In particular, 11 gauging stations located along the main river network of the Mississippi River basin have been selected to represent the spatial distribution of river discharge over the basin. The location of these gauging stations along with relevant characteristics (e.g., the upstream basin area, the mean annual river discharge and the presence of upstream dams) are summarized in Table 1. Mean annual river discharge ranges from 141 to 17'500 $m^3$/s, and 3 of 11 gages are located downstream of big dams (Lehner et al., 2011). In particular, gages 1, 2 and 5 are located downstream of Garrison (the fifth-largest earthen dam in the world), Gavins Point and Kanopolis dams, respectively (see Figure 1a and Table 1). The related reservoirs have a maximum storage of $29.383 \times 10^9$ $m^3$, $0.607 \times 10^9$ $m^3$, and $1.058 \times 10^9$ $m^3$, respectively.

**3.2 Satellite Products**

Satellite products include observations of precipitation, soil moisture and TWSA.

The satellite precipitation dataset used in this study is the Multi-satellite Precipitation Analysis 3B42 Version 7 (her after referred to as TMPA) estimate produced by the National Aeronautics and Space Administration (NASA) as the 0.25°×0.25° quasi-global (50°S-50°N) gridded dataset. The TMPA is a gauged-corrected satellite product, with a latency period of two months, available at 3h sampling interval from 1998 to present. Major details about the *P* dataset, downloadable from http://pmm.nasa.gov/data-access/downloads/trmm, can be found in Huffman et al. (2007).

Soil moisture data have been taken from the European Space Agency Climate Change Initiative (ESA CCI) Soil Moisture project (https://esa-soilmoisture-cci.org/) that provides a surface soil moisture product (referred to first 2–3 cm of soil) continuously updated in terms of spatial-temporal coverage, sensors and retrieval algorithms (Dorigo et al., 2017). In this study, the daily combined ESA CCI soil moisture product v4.2 is used. It is available at global scale with a grid spacing of 0.25°, for the period 1978 to present.

TWSA have been obtained from the Gravity Recovery And Climate Experiment (GRACE) satellite
mission. Here we employ the NASA Goddard Space Flight Center (GSFC) global mascon model,
i.e., Release v02.4, (Luthcke et al. 2013). It has been produced based on the mass concentration
(mascon) approach. The model provides surface mass densities on a monthly basis. Each monthly
solution represents the average of surface mass densities within the month, referenced at the middle
of the corresponding month. The model has been developed directly from GRACE level-1b K-Band
Ranging (KBR) data. It is computed and delivered as surface mass densities per patch over blocks of
approximately $1°\times1°$ or about 12'000 km$^2$. Although the mascon size is smaller than the inherent
spatial resolution of GRACE of about $2.5°\times2.5°$ or 64'000 km$^2$ (Vishwakarma et al., 2018), the model
exhibits a relatively high spatial resolution. This is attributed to a statistically optimal Wiener
filtering, which uses signal and noise full covariance matrices. This allows the filter to fine tune the
smoothing in line with the signal-to-noise ratio in different areas. That is, the less smoothing, the
higher signal-to-noise ratio in a particular area and vice versa. This ensures that the filtering is
minimal and aggressive smoothing is avoided when unnecessary. Further details of such a filter can
be found in Klees et al. (2008). Importantly, the coloured noise characteristic of KBR data was taken
in to account when compiling the GRACE model, which has allowed for a reliable computation of
the aforementioned noise full covariance matrices. The coloured noise characteristic of KBR data
was taken into account when compiling the model, which has allowed for a reliable computation of
these noise and signal covariance matrices. They play a crucial role when filtering and allow a higher
spatial resolution compared to commonly applied GRACE filtering methods such as Gaussian
smoothing and/or destriping filters. The GRACE data used here are available from January 2003 to
July 2016, which suffices to demonstrate the STREAM capabilities. With its successor mission
GRACE Follow-On (GRACE-FO), launched early 2018, the time series of time-variable gravity has
reached a nearly uninterrupted time span of about 20 years, thus allowing a continued and operational
use of STREAM. The existing interruptions, short ones due to mission operations or technical
failures, but also the one-year gap between GRACE and GRACE-FO can be dealt with in various
ways, e.g. by data driven gap filling (Yi and Sneeuw, 2021).

**3.3 Runoff Verification Data**

To establish the quality of the STREAM model in runoff simulation, monthly runoff data obtained
from the Global Runoff Reconstruction (GRUN_v1, https://doi.org/10.3929/ethz-b-000324386) have
been used for comparison. The GRUN dataset (Ghiggi et al., 2019) is a global monthly runoff dataset
derived through the use of a machine learning algorithm trained with in situ river discharge
observations of relatively small catchments ($<2500$ km$^2$) and gridded precipitation and temperature
derived from the Global Soil Wetness Project Phase 3 (GSWP3) dataset (Kim et al., 2017). The
dataset covers the period from 1902 to 2014 and it is provided on a 0.5° ×0.5° regular grid.

**4. METHOD**

**4.1 STREAM Model: the Concept**

The STREAM model conceives river discharge as a combination of hydrological responses operating
at diverse time scales (Blöschl et al., 2013; Rakovec et al., 2016). In particular, river discharge can
be considered made up of a *slow-flow* component, produced as outflow of the groundwater storage
and of a *quick-flow* component, i.e. mainly related to the surface and shallow-subsurface runoff
components (Hu and Li, 2018).
While the high spatial and temporal variability of precipitation and the highly changing land cover
spatial distribution significantly impact the variability of the *quick-flow* river discharge component
(with scales ranging from hours to days and metres to kilometres depending on the basin size), *slow-*
*flow* river discharge reacts to precipitation inputs more slowly as water infiltrates, is stored, mixed
and is eventually released in times spanning from weeks to months. Therefore, the two components
can be estimated by relying upon two different approaches that involve different types of
observations. Based on that, within the STREAM model, satellite soil moisture, precipitation and
TWSA will be used for deriving river discharge and runoff estimates. The first two variables are used

as proxy of the *quick-flow* river discharge component while TWSA is exploited for obtaining its complementary part, i.e., the *slow-flow* river discharge component. Firstly, we exploit the role of the soil moisture in determining the response of the catchment to the precipitation inputs, which have been soundly demonstrated in more than ten years of literature studies (see e.g., Brocca et al., 2017 for a comprehensive discussion on the topic). Secondly, we consider the important role of total water storage in determining the *slow-flow* river discharge component as modelled in several hydrological models (e.g., Sneeuw et al., 2014).

It is worth noting that modeling the *quick-flow* and *slow-flow* river discharge components independently has been largely applied and tested in recent and past studies, e.g., for the estimation of the flow duration curve (see e.g, Botter et al., 2007a, b; Yokoo and Sivapalan 2011; Muneepeerakul et al., 2010; Ghotbi et al., 2020).

**4.2 STREAM Model**

The STREAM model is a semi-distributed conceptual hydrological model that uses gridded satellite-derived inputs of precipitation, soil moisture, TWSA and air temperature to estimate daily values of gridded runoff and river discharge time series at select basin outlets.

To set up the model, the catchment is divided into $b$ sub-catchments, each one representing either a tributary draining area with outlet along the main channel or an area draining directly into the main channel (see Figure 2). Each sub-catchment, assumed homogeneous, is further divided into an array $N_b$ of individual cells assumed as the unit basis for the runoff generation. Note that the number $N_b$ differs for each sub-catchment as, for a fixed cell grid size, it varies with the sub-catchment area. Once estimated at cell scale and aggregated at the sub-basin scale (see section 4.2.1 for details), the runoff is routed at each sub-catchment outlet (see section 4.2.2) and then transferred through the channels and the rivers for the computation of the river discharge at intermediate outlets or at the outlet of the entire basin (see section 4.2.3).

Based on that, hereinafter we refer to river discharge, $Q$, to indicate the amount of water passing a
particular point of a river (in $m^3 \, s^{-1}$) whereas runoff, $R$, is regarded as the depth of water produced
from a drainage area during a particular time interval (in mm). The difference between the two
quantities is related to the routing processes that allow to transform the runoff into river discharge.
**4.2.1 Runoff generation at cell scale**
The soil zone of each cell $i$ of the basin is divided into two layers, the upper and lower soil storages
allowing to model the related runoff responses, $R_{q,i}$ [mm] and $R_{s,i}$ [mm], as illustrated in Figure 2b.
The upper cell storage receives inputs from precipitation ($P_i$), released through a snow module
(Cislaghi et al., 2020) as rainfall ($r_i$) or stored as snow water equivalent ($SWE_i$) within the snowpack
and on the glaciers. In particular, according to Cislaghi et al. (2020), $SWE_i$ is modelled by using as
input air temperature ($T_{air,\,i}$) and a degree-day coefficient, $C_m$, to be estimated by calibration.
Once precipitation is partitioned by the snow model, the rainfall output $r_i$ contributes to $R_{q,i}$ while the
$SWE_i$ (like other fluxes contributing to modify the soil water content into $Su$) is neglected as already
considered in the satellite TWSA. Therefore, the first key point of the STREAM model is that the
water content in the upper storage of soil zone, $Su$ (Figure 2b), is directly provided by the satellite
soil moisture observations and the loss processes like percolation or evaporation do not need to be
explicitly modelled to estimate the evolution in time of soil moisture. Consequently, for each cell $i$,
$R_{q,i}$ can be computed following the formulation proposed by Georgakakos and Baumer (1996), as in
equation (1):
$$R_{q,i}\,(t) = r_i(t)\,SWI_i(t,T)^\alpha \qquad\qquad\qquad\qquad (1)$$
where:
-    $t$ [days] represents the time;
-    $r_i$ [mm] is the rainfall, obtained as an output from the snow module;
-    $SWI_i$ [-] is the Soil Water Index (Wagner et al., 1999), i.e., the root-zone soil moisture product

referred to the first layer of the model (representative of the first 5–30 cm of soil), derived by the

surface satellite soil moisture product, $\theta_i$, by applying the exponential filtering approach in its

recursive formulation (Albergel et al., 2009):

$$SWI_{i,n} = SWI_{i,n-1} + K_n(\theta_i(t_n) - SWI_{i,n-1})$$            *(2)*

with the gain $K_n$ at the time $t_n$ given by:

$$K_n = \frac{K_{n-1}}{K_{n-1} + e^{\left(\frac{t_n - t_{n-1}}{T}\right)}}$$            *(3)*

-    $T$ [days] is a parameter, named characteristic time length, that characterizes the temporal variation

of soil moisture within the root-zone profile and the gain $K_n$ ranges between 0 and 1;

-    $\alpha$[-] is a coefficient linked to the non-linearity of the infiltration process and it considers the

characteristics of the soil;

-    for the initialization of the filter $K_1 = 1$ and $SWI_1 = \theta(t_1)$.
The second key point of STREAM model concerns the estimation of $R_{s,i}$, i.e., the *slow-runoff* response
related to the lower storage of the soil zone. The hypothesis here, shared also with other studies (e.g.,
Rakovec et al., 2016), is that the dynamic of $R_s$ can be represented by the monthly TWSA data. Indeed,
the time scale of $R_s$ is typically in the range of seasons to years and it can be assumed almost
independent of the water that is contained in the upper storage. For that, for each cell $i$, $R_{s,i}$ can be
computed following the formulation proposed by Famiglietti and Wood (1994), through equation (4)
as follows:
$$R_{s,i}(t) = \beta \, (TWSA_i^*(t))^m$$            *(4)*
where:
-    $TWSA_i^*$ [-] is the TWSA estimated by GRACE over the cell $i$ normalized by its minimum and

maximum values. The assumption behind this equation is that TWSA can be assumed as a proxy

of the evolution in time of the $Sl$, i.e., the water amount in the lower storage of the soil zone.

-    $\beta$ [mm h$^{-1}$] and $m$ [-] are two parameters describing the nonlinearity between lower storage runoff

component and $TWSA^*$.

Note that we made the hypothesis that soil moisture and TWSA observations are independent
(whereas in reality soil moisture can be responsible both for the generation of $R_q$ (mainly) and for the
$R_s$ contribution) given the different temporal (and spatial) scales at which the upper and lower runoff
responses act.
By neglecting any lateral flow, the runoff responses at cell scale are averaged at sub-catchment scale
to obtain $b$ runoff responses, one for each sub-catchment. Specifically, by considering $N_b$ cells for
each sub-catchment, the following equation are used:
$$R_{q,b}(t) = \frac{\sum_{i=1}^{N_b} R_{q,i}(t)}{N_b} \hspace{5cm} (5)$$
$$R_{s,b}(t) = \frac{\sum_{i=1}^{N_b} R_{s,i}(t)}{N_b} \hspace{5cm} (6)$$
**4.2.2 Sub-catchment river discharge calculation**
For each sub-catchment $b$, the runoff component $R_{q,b}$ is routed to its outlet by the Geomorphological
Instantaneous Unit Hydro-graph (GIUH, Gupta et al., 1980) for tributary draining areas or through a
linear reservoir approach (Nash, 1957) for directly draining areas. The $R_{s,b}$ runoff component is
transferred to the sub-catchment outlet by a linear reservoir approach. These processes are controlled
by a parameter lag time, $L$ [days], evaluated as (Corradini et al., 2002):
$$L = \gamma 1.19 \, A_b^{0.33} \hspace{6cm} (7)$$
where $A_b$ [km$^2$] is the sub-catchment area and $\gamma$ [-] is a parameter to be calibrated.
By routing the $R_{q,b}$ and $R_{s,b}$ components the *quick-flow*, $Q_{q,b}$ [m³/s], and the *slow-flow, $Q_{s,b}$* [m³/s]
river discharge components at each sub-catchment outlet are obtained (see Figure 2c).
**4.2.3 River discharge routing through river networks**
A diffusive linear approach (controlled by the parameters $C$ [km h⁻¹] and $D$ [km² h⁻¹], i.e., Celerity
and Diffusivity, Troutman and Karlinger, 1985) is applied to route the two river discharge
components, $Q_{q,b}$ and $Q_{s,b}$ trough the river network from the sub-catchment outlet to intermediate
outlets along the river or to the outlet of the entire basin (Brocca et al., 2011). In this way the *quick-*
*flow, $Q_q$* [m³/s], and the *slow-flow, $Q_s$* [m³/s] river discharge components at the catchment outlet are
obtained (see Figure 2d).
**4.3 STREAM Parameters**
The STREAM model uses 8 calibration parameters for each sub-catchment $b$ into which the entire
basin is divided. Among these parameters, 5 control the runoff generation process ($\boldsymbol{\alpha}$, $\boldsymbol{T}$, $\boldsymbol{\beta}$ , $\boldsymbol{m}$, $\boldsymbol{C}_{\mathrm{M}}$)
and 3 the routing component and therefore the streamflow dynamics ($\boldsymbol{\gamma}$, $C$ and $D$). The parameter
values determined within the feasible parameter space (See Table Appendix A for more details), are
calibrated by maximizing the Kling-Gupta Efficiency index ($\boldsymbol{KGE}$, Gupta et al., 2009; Kling et al.,
2012, see section 5.1 for more details) between observed and modelled river discharge. For model
calibration, a standard gradient-based automatic optimisation method (Bober 2013) was used.
**5.  EXPERIMENTAL DESIGN**
**5.1 Modelling Setup for Mississippi River Basin**
The modelling setup is carried out in three steps (Figure 3):
1. *Sub-catchment delineation.* The TopoToolbox (https://topotoolbox.wordpress.com/), a tool
developed in Matlab by Schwanghart and Kuhn (2010), and the SHuttle Elevation Derivatives at
multiple Scales (HydroSHED, https://www.hydrosheds.org/) DEM of the basin at the 3″ resolution
(nearly 90 m at the equator) have been used to derive flow directions, to extract the stream network
and to delineate the drainage basins over the Mississippi River basin. In particular, by considering
only rivers with order greater than 3 (according to the Horton-Strahler rules, Horton, 1945; Strahler,
1952), the Mississippi watershed has been divided into 53 sub-catchments as illustrated in Figure 1a.
Blue lines in the figure illustrate the river network pathway connecting the sub-catchments, red dots
indicate the location of the 11 river discharge gauging stations selected for the study area.
It has to be specified that the step of sub-basin delineation could be accomplished through tools
different from the TopoToolbox. For instance, it could be used the free Qgis software downloadable
at https://www.qgis.org/it/site/forusers/download.html, following the instruction to perform the
hydrological                        analysis                        as                        in
https://docs.qgis.org/3.16/en/docs/training_manual/processing/hydro.html?highlight=hydrological%
20analysis.
2. *Extraction of input data.* Precipitation, air temperature, soil moisture and TWSA datasets data have
to be extracted for each sub-catchment of the study area. If characterized by different spatial/temporal
resolution, these datasets need to be resampled over a common spatial grid/temporal time step prior
to be used as input into the model.
To run the STREAM model over the Mississippi river basin, input data have been resampled over the
precipitation spatial grid at 0.25° resolution through a bilinear interpolation. Concerning the temporal
scale, air temperature, soil moisture and precipitation data are available at daily time step, while
monthly TWSA data have been linearly interpolated at daily time step. For each of the 53 Mississippi
sub-catchment, the resampled precipitation, soil moisture, air temperature and TWSA data have been
extracted (see Figure 1b and1c).
3. *STREAM model calibration.* In situ river discharge data are used as reference data for the
calibration of STREAM model. For Mississippi, the STREAM model has been calibrated at five
gauging stations, i.e., the stations 4, 6, 9, 11 and 10. This allowed to identify five sets of STREAM
parameters attributed to each catchment according to the river network pathway illustrated in Figure
1a. This means that, for example, to the sub-catchments labelled as 1, 2, 5 to 15, 17, 22, 23, and 30
contributing to the gauging station 4 are attributed the parameter set obtained by calibrating the model
against river discharge data observed at station 4; to the sub-catchments 31, 37, 38 and 41 contributing
to gauging station 6 are attributed the parameter set obtained by calibrating the model with respect to
gauging station 6 and so on. Consequently, the sub-catchments highlighted with the same colour in
Figure 1a are assigned the same model parameters, i.e. the parameters that allow to reproduce the
river discharge data observed at the related gage.
Once calibrated, the STREAM model has been run to provide continuous daily runoff and river
discharge time series, over each grid pixel and at the outlet section of each sub-catchment,
respectively. By considering the spatial/temporal availability of both in situ and satellite observations,
the entire analysis period covers the maximum common observation period, i.e., from January 2003
to July 2016 at daily time scale. To establish the goodness-of-fit of the model, the modelled river
discharge and runoff timeseries are compared against in situ river discharge and modelled runoff data.
**5.2 Model Evaluation Criteria and Performance Metrics**
The model has been run over a 13.5-year period split into two sub periods: the first 8 years, from
January 2003 to December 2010, are used to calibrate the model. The model is validated, as described
below over the remaining 5.5 years (January 2011 - July 2016).
In particular, three different validation schemes have been adopted to assess the robustness of the
STREAM model:
1.  internal validation aimed to test the plausibility of both the model structure and the parameter set

in providing reliable estimates of the hydrological variables against which the model is calibrated.

For this purpose, a comparison between observed and modelled river discharge time series on the

gauging stations used for model calibration has been carried out for both the calibration and

validation sub periods;

2.  cross-validation testing the goodness of the model structure and the calibrated model parameters

to predict hydrological variables at locations not considered in the calibration phase. In this

respect, the cross-validation has been carried out by comparing observed and modelled river
discharge time series in gauging stations not considered during the calibration phase;
3. external validation aimed to test the capability of the model "*to get the right answers for the right*
*reasons*" (Kirchner 2006). The rationale behind this concept is that the hydrological models are
today highly performing and able to reproduce a lot of hydrological variables. For that, the model
performances should not only be evaluated against observed river discharge, but complementary
datasets representing internal hydrologic states and fluxes (e.g., soil moisture, evapotranspiration,
runoff etc) should be considered. As runoff is a secondary product of the STREAM model,
obtained indirectly from the calibration of the river discharge (basin-integrated runoff), the
comparison in terms of runoff can be considered as a further external validation of the model.
Runoff, differently from river discharge, cannot be directly measured. It is generally modelled
through land surface or hydrological models. Its validation requires a comparison against
modelled data that, however, suffer from uncertainties (Beck et al., 2017). Based on that, in this
study the GRUN runoff dataset described in the section 3.3 has been used for a qualitative
comparison.
**5.3 Performance Metrics**
To measure the goodness-of-fit between modelled and observed river discharge data three
performance scores have been used:
● the root mean square error relative to the mean, $RRMSE$:
$$RRMSE = \frac{\sqrt{\frac{1}{n}\sum_{j=1}^{n}(Qmod_j - Q_{obs_j})^2}}{\frac{1}{n}\sum_{j=1}^{n}(Q_{obs_j})} \quad (8)$$
where $Q_{obs}$ and $Q_{mod}$ are the observed and modelled river discharge time series of length $n$. $RRMSE$
values range from 0 to +∞, the lower the $RRMSE$ , the better the agreement between observed and
modelled data.
● the Pearson correlation coefficient, $rho$, measuring the linear relationship between two variables:
$$rho = \frac{\sum_{j=1}^{n}(Qmod_j - \overline{Q_{mod}})(Qobs_j - \overline{Q_{obs}})}{\sqrt{\sum_{j=1}^{n}(Qmod_i - \overline{Q_{mod}})^2(Qobs_j - \overline{Q_{obs}})^2}} \qquad (9)$$
where $\overline{Q_{obs}}$ and $\overline{Q_{mod}}$ represent the mean values of $Q_{obs}$ and $Q_{mod}$, respectively. The values of $rho$
range between $-1$ and 1; higher values of R indicate a better agreement between observed and
modelled data.
● the Kling-Gupta efficiency index ($KGE$, Gupta et al., 2009), which provides direct assessment of
four aspects of river discharge time series, namely shape, timing, water balance and variability.
It is defined as follows:
$$KGE = 1 - \sqrt{(rho - 1)^2 + (\delta - 1)^2 + (\varepsilon - 1)^2} \qquad (10)$$
where $\delta$ is the relative variability and $\varepsilon$ the bias normalized by the standard deviation between
observed and modelled river discharge. The $KGE$ values range between $-\infty$ and 1; the higher the $KGE$
the better is the agreement between observed and modelled data. Simulations characterized by values
of $KGE$ in the range -0.41 and 1 can be assumed as reliable; values of $KGE$ greater than 0.5 have been
assumed good with respect to their ability to reproduce observed time series (Thiemig et al., 2013).
**5.4 STREAM sensitivity analysis**
To investigate how the variation of the STREAM parameters influences the variation of the STREAM
model outputs, a global sensitivity analysis has been carried out. Specifically, the Variance-Based
sensitivity analysis (VBSA, Sobol 1993) implemented into the Sensitivity Analysis For Everybody
toolbox (SAFE, Pianosi et al., 2015, https://www.safetoolbox.info/) has been applied. VBSA relies
on the variance decomposition and consists of assessing the contributions to the variance of the model
output from variations in the parameters. In this study, we use as sensitivity index the first-order (main
effect) index, which measures the variance contribution from variations in an individual input factor
alone (i.e., excluding interactions with other factors) and the total sensitivity indices, which measure
the total contribution of a single input factor or a group of inputs including interactions with all other
inputs. The following steps were carried out to execute the VBSA. Firstly, the locality-sensitive
hashing (LSH) technique was used to generate 15000 samples from the model parameter space (see
Table 1A). Previous hydrological studies (e.g., Tang et al., 2007) recommend the LHS sampling
method for its sampling efficiency. Secondly, 15000 STREAM model runs were executed and the
corresponding $KGE$ values (11x15000 values, one for each gauging station for each run) were
retained. Thirdly, the parameters and the 15000 $KGE$ samples were used in the SAFE toolbox to
compute the sensitivity indices.
For major details on the workflow needed to implement the VBSA the reader is referred to Noacco
et al. (2020).
**6.   RESULTS**
The testing and validation of the STREAM model is presented and discussed in this section according
to the scheme illustrated in section 5.2.
**6.1 Internal Validation**
The performance of the STREAM model over the gauging stations used for calibration is illustrated
in Figure 4 and summarized in Table 2. Figure 4 shows observed and modelled river discharge time
series over the whole study period (2003-2016); in Table 2 the performance scores are evaluated
separately for the calibration and validation sub periods. It is worth noting that the model accurately
predicts the observed river discharge data and is able to give the "right answer" with good modelling
performances. Score values of $KGE$ and $rho$ over the calibration period are higher than 0.78 for all
the calibrated gauging stations; $RRMSE$ is lower than 45% for all the calibrated gauging stations
except for station 9, where it rises up to 66%. The performances remain good even if they are
evaluated over the validation period or the entire study period as indicated by the scores on the top of
each plot of Figure 4.
**6.2 Cross-validation**
The cross-validation has been carried out over the six gauging stations illustrated in Figure 5 not used
in the calibration step. The performance scores on the top of each plot refer to the entire study periods;
the scores split for calibration and validation periods are reported in Table 2. For some river discharge
gauging stations the performance is quite low (see, e.g., gauging station 1, 2 and 5) whereas for others
the model is able to estimate river discharge data quite accurately (e.g., 7 and 8). In particular, for the
gauging stations 1 and 2 even if $KGE$ reaches values equal to 0.39 and 0.46 for the whole period,
respectively, there is not a good agreement between observed and modelled river discharge and the
$rho$ score is lower than 0.56 for both the stations. The worst performance is obtained over the gauging
station 5, with negative $KGE$ and low $rho$ values. These results are certainly influenced by the
presence of large dams located upstream to these stations (i.e., Garrison, Gavins Point and Kanopolis
dams, see Table 1) which have a strong impact on river discharge: the model, not having a specific
module for modelling reservoirs, is not able to accurately reproduce the dynamics of river discharge
over regulated river stations. Positive $KGE$ values are obtained over the gauging stations 3, 7 and 8.
In particular, over the gauging station 3 the STREAM model overestimates the observed river
discharge due the presence of large dams along the Missouri river, over the Great Plains region. This
area is well known from other large-scale hydrological models (e. g., ParFlow-CLM and WRF-
Hydro) to be an area with very low performances in terms of river discharge modelling (O'Neill et
al., 2020, Tijerina et al., 2021).
Over the gauging station 7, located over the Rock river, a relatively small tributary of the Mississippi
river (see Table 1), the STREAM model overestimation has to be attributed to: 1) the different
characteristics of the Rock river basin with respect to the entire basin closed to station 6 where the
model has been calibrated (see Figure 1a); 2) the small size of the Rock river basin (23'000 km$^2$ , if
compared with GRACE resolution, 160'000 km$^2$) for which the model accuracy is expect to be lower.
Conversely, the performances over the gauging station 8, whose parameters have been set equal to
the ones of gauging station 10, are quite high ($KGE$ equal to 0.71, 0.81 and 0.78 for the entire, the
calibration and the validation period, respectively; $rho$ equal to 0.82, 0.84 and 0.83 for the entire,
calibration and validation periods, respectively). This outcome demonstrates that under some
circumstances, the STREAM model can be used to estimate river discharge in basins not calibrated
over, especially those without upstream dams and with comparable size and land cover.
On overall, the cross-validation results suggest that the performances of STREAM model, as any
hydrological model calibrated against observed data, decrease over the gauging stations not used for
the calibration raising doubts about the robustness of model parameters and whether it is actually
possible to transfer model parameters from one river section to another with different inter-basin
characteristics. A more in-depth investigation about the model calibration procedure, with special
focus on the regionalization of the model parameters, should be carried out but this topic is beyond
the scope of the manuscript.
**6.3 External Validation**
For the external validation, the monthly runoff time series provided by the GRUN datasets have been
compared against the ones computed by the STREAM model. For that, STREAM daily runoff time
series have been aggregated at monthly scale and re-gridded at the same spatial resolution of the
GRUN dataset (0.5°). The comparison is illustrated in Figure 6 for the common period 2003−2014.
Although the two datasets consider different precipitation inputs, the two models agree in identifying
two distinct zones in terms of runoff, i.e., the western dry and the eastern wet area. These two distinct
zones can be clearly identified also in the GSWP3 and TMPA 3B42 V7 precipitation maps (see Figure
A1) used as input in GRUN and STREAM, respectively, stressing that STREAM runoff output is
correctly driven by the input data. However, likely due to the calibration procedure, the STREAM
runoff map appears patchier with respect to GRUN and discontinuities along the sub-basin boundaries
(identified in Figure 1a) can be noted. This should be ascribed to the automatic calibration procedure
of the model that, differently from other calibration techniques (e. g., regionalization procedures),
does not consider the basin physical attributes like soil, vegetation, and geological properties that
govern spatial dynamics of hydrological processes. This calibration procedure can generate sharp
discontinuities even for neighbouring sub-catchments individually calibrated. It leads to
discontinuities in model parameter values and consequently in the modelled hydrological variable
(runoff).
**6.4 Sensitivity analysis results**
The results of the VBSA, are illustrated in Figure 7a in terms of main effect indices and in Figure 7b
in terms of total effect. Specifically, the figure refers to Vicksburg station but similar results have
been obtained for all the 11 gauging stations in the Mississippi basin. By looking at Figure 7, we
observe that the model parameters most influencing the model response are $\beta$ and $m$, i.e., the two
parameters controlling the *slow-flow* runoff response of the lower soil storage. In particular, the total
effect sensitivity index of these two parameters is higher than the main effect sensitivity index. This
means that these two parameters have an effect on the model output not only through their individual
variations but also through interactions with other parameters. Instead, the other five parameters ($\alpha$,
$T, \gamma, C, D$ and $C_m$) have low main and total effect indices, and consequently, these parameters have
a small effect, both direct and through interactions, on model response. Among these, only the
$\alpha$ parameter shows a slightly high main and total effect sensitivity indices.
This outcome is very important as it allows to clearly distinguish model parameters which values
should be carefully determined when calibrating the model ($\beta$ and $m$ and partially $\alpha$) from the least
sensitive ($T, \gamma, C, D$ and $C_m$) which values could be set values within the model parameters' range of
variability and then excluded during the calibration phase.
**7.  DISCUSSION**
In the previous sections, the ability of the STREAM model to estimate river discharge and runoff
time series has been presented. In particular, Figures 4, 5 and 6 demonstrate that satellite observations
of precipitation, soil moisture and total water storage anomalies can provide accurate daily river
discharge estimates for near-natural large basins (absence of upstream dams), and for basins with
draining area greater than 160'000 km$^2$ (see section 6.2), i.e., at spatial/temporal resolution greater
than the ones of the TWSA input data (monthly, 160'000 km$^2$). This is an important result of the
study as it demonstrates, on one hand, that the model structure is appropriate with respect to the data
used as input and, on the other hand, the great value of information contained into TWSA data that,
even if characterized by limited spatial/temporal resolution, can be used to estimate runoff and river
discharge at basin scale. This finding has been also confirmed by a preliminary sensitivity analysis in
which the STREAM model has been run with different hydrological inputs of precipitation, soil
moisture and total water storage anomaly (not shown here for brevity). In particular, by running the
STREAM model with different input configurations (e.g., by using TMPA 3B42 V7 or CPC data for
precipitation, ESA CCI or Advanced SCATterometer (ASCAT) data for soil moisture, TWSA or ESA
CCI soil moisture data to model the slow-flow river discharge component), we found that STREAM
results are more sensitive to soil moisture data rather than to precipitation input. In addition, by
running STREAM model with soil moisture data as input to model the slow-flow river discharge
component (i.e. without using TWSA data) we found a deterioration of the model results. This
outcome along with the one obtained in the section 6.3, demonstrating the high sensitivity of the
model parameters related to *slow-flow* river discharge component, confirm the paramount role of
TWSA in estimating river discharge. In this respect, the availability of GRACE data up to July 2016
could represent an issue for the model application beyond that date. However, the GRACE-FO along
with the numerous literature studies devoted to fill the GRACE data gap between GRACE and
GRACE-FO (see e.g., Landerer et al., 2020 or Yi and Sneeuw, 2021), can provide the needed data to
extend the STREAM model application up to present. Further developments in this direction are
expected with the ESA's Next Generation Gravity Mission (NGGM), a candidate Mission of
Opportunity for ESA–NASA cooperation in the frame of the Mass Change and Geosciences
International Constellation (MAGIC) that will enable long-term monitoring of the temporal variations
of Earth's gravity field at relatively high temporal (down to 3 days) and increased spatial resolutions
(up to 100 km). This implies also that time series of GRACE and GRACE-FO can be extended
towards a climate series (Massotti et al., 2021).

By looking at technical reviews of large-scale hydrological models (e.g., Sood and Smakhtin, 2015, Kauffeldt et al., 2016), it can be noted there are many established models, similar in objective and limitations to STREAM model, already existing with support and user base (e.g., among others, Community Land Model, CLM, Oleson et al., 2013; European Hydrological Predictions for the Environment, E-HYPE, Lindström et al., 2010; H08, Hanasaki et al., 2008, PCR-GLOBWB, van Beek and Bierkens, 2008; Water – a Global Assessment and Prognosis WaterGAP, Alcamo et al., 2003; ParFlow–CLM, Maxwell et al., 2015; WRF-Hydro, Gochis et al., 2018; Precipitation-Runoff Modeling System, PRMS; Markstrom et al., 2015). Some of them, e.g., ParFlow-CLM, WRF-Hydro or PRMS have been specifically configured across the continental United States and showed good capability to reproduce observed streamflow data over the Mississippi river basin with performances decreased throughout the Great Plains (O'Neill et al., 2020, Tijerina et al., 2021) which is consistent with the results we obtained with the STREAM model. However, with respect to classical hydrological and land surface models, STREAM is based on a new concept for estimating runoff and river discharge which relies on the almost exclusive use of satellite observations, and, a simplification of the processes being modelled.

This approach brings several advantages: 1) satellite data implicitly consider the human impact on the water cycle observing some processes, such as irrigation application or groundwater withdrawals, that are affected by large uncertainty in classical hydrological models, 2) the satellite technology grows quickly and hence it is expected that the spatial/temporal resolution and accuracy of satellite products will be improved in the near future (e.g., 1 km resolution from new satellite soil moisture products and the next generation gravity mission); the STREAM model is able to fully exploit such improvements; 3) STREAM model models only the most important processes affecting the generation of runoff, and considers only the most important variables as input (precipitation, surface soil moisture and groundwater storage). In other words, the model does not need to parametrize processes, such as evapotranspiration and percolation and therefore it is an independent modelling

approach for simulating runoff and river discharge that can be also exploited for benchmarking and
improving classical land surface and hydrological models.
**7.1 Strengths and limitations of STREAM model**
Hereinafter, the strengths and the main limitations of the STREAM model are discussed.
Among the strengths of the STREAM model it is worth highlighting:
**Simplicity**. The STREAM model structure: 1) limits the input data required. Only precipitation, air
temperature, soil moisture and TWSA data are needed as input whereas LSM/GHMs require many
additional inputs such as wind speed, shortwave and longwave radiation, pressure and relative
humidity; 2) limits and simplifies the processes to be modelled for runoff and river discharge
simulation. Processes like evapotranspiration or percolation, are not modelled therefore avoiding the
need of using sophisticated and highly parameterized equations (e.g., Penman-Monteith for
evapotranspiration, Allen et al.,1998); 3) limits the number of parameters (only 8 parameters have to
be calibrated) thus simplifying the calibration procedure and potentially reduces the model
uncertainties related to the estimation of parameter values.
In particular, the STREAM model is even simpler than the classical semi-distributed conceptual
hydrological models available in literature. As an example, for the comparison we could refer to the
Hydrologiska Byråns Vattenbalansavdelning model (HBV, Bergström 1995) or to the Hydrologic
Engineering Center – Hydrologic Modeling System (HEC-HMS, Feldman, 2000). HBV model counts
14 parameters to be calibrated and needs precipitation, air temperature and potential
evapotranspiration as input data. Similar input data are required for HEC-HMS which counts 23
parameters. Both the models, uses conceptual equations to estimate the soil losses and to model the
soil water storage.
**Versatility**. The STREAM model is a versatile model suitable for daily runoff and river discharge
estimation over sub-basins characterized by different physiographic/climatic characteristics (see e.g.,
the outcomes obtained for the gages 9 and 11 located in the driest and wetter part of the Mississippi
basin). This aspect is paramount as it gives an insight about the potential of the model to be extended

at the global scale. Moreover, the model can be easily adapted to ingest input data with spatial/temporal resolution different from the one tested in this study (0.25°/daily). For instance, satellite missions with higher space/time resolution (e.g., GPM Final Run, ASCAT and NGGM-MAGIC) or near-real time products (e.g., GPM Early Run, EUMETSAT H16, GRACE European Gravity Service for Improved Emergency Management, EGSIEM GRACE data Jäggi et al., 2019) could be considered.

Additionally, the STREAM model shows highly flexibility as: 1) it can accommodate application domains comprising single or multiple basins of any size; and 2) the sub-catchment delineation procedure can be easily adapted to introduce intermediate outlets along the river in correspondence of gages with available observed river discharge data, useful for model calibration.

**Low computational cost.** Due to its simplicity and the limited number of parameters to be calibrated, the computational effort for the STREAM model is very limited (model runs requiring seconds to minutes). For instance, a run of the STREAM model over the presented case study takes less than 2 seconds on a machine with 16 GB RAM and 4 Core.

However, some limitations have to be acknowledged for the current version of the STREAM model:

**Presence of reservoir, diversion, dams or flood plain**. As the STREAM model does not explicitly consider the presence of discontinuity elements along the river network (e. g, reservoir, dam or floodplain), river discharge estimates obtained for gauging stations located downstream of such elements might be inaccurate (see, e.g., gauging stations 1 and 2 in Figure 5).

**Snow modelling**. A potential limitation of the current version of the STREAM model is related to the rain/snow differentiation, based on the degree-day coefficient. A different scheme based e.g., on the wet bulb temperature like in IMERG (Wang et al., 2019; Arabzadeh and Behrangi, 2021), could be investigated in future developments.

**Need of in situ data for model calibration and robustness of model parameters**. As discussed in the results section, the parameter values of the STREAM model are set through an automatic calibration procedure aimed at minimizing the differences between modelled and observed river

discharge. The main drawbacks of this parameterization technique are a poor predictability of state
variables and fluxes at locations and periods not considered in the calibration, and the presence of
sharp discontinuities along sub-basin boundaries in state flux and parameter fields (e.g., Merz and
Blöschl, 2004). To overcome these issues, several regionalization procedures, as for instance
summarized in Cislaghi et al. (2020), could be conveniently applied to transfer model parameters
from hydrologically similar catchments to a catchment of interest. In particular, the regionalization
of model parameters could allow to, firstly, estimate river discharge and runoff time series over
ungauged basins overcoming the need of river discharge data recorded from in–situ networks,
secondly, estimate the model parameter values through a physically consistent approach, linking them
to the characteristics of the basins and, thirdly, solve the problem of discontinuities in the model
parameters, avoiding to obtain patchy unrealistic runoff maps. As this aspect requires additional
investigations and it is beyond the paper purpose, it will not be tackled here.
**8.  CONCLUSIONS**
This study presents a new conceptual hydrological model, STREAM, for runoff and river discharge
estimation. By using as input satellite data of precipitation, soil moisture and total water storage
anomalies, the model has been able to provide accurate daily river discharge and runoff estimates at
the outlet river section and the inner river sections and over a 0.25°×0.25° spatial grid of the
Mississippi river basin. In particular, the model is suitable to reproduce:
1. river discharge time series over the calibrated river section with good performances both in
calibration and validation periods;
2. river discharge time series over river sections not used for calibration and not located downstream
dams or reservoirs;
3. runoff time series with a quite good agreement with respect to the well-established GRUN
observational-based dataset used for comparison.
The integration of observations of soil moisture, precipitation and total water storage anomalies is a
first alternative method for river discharge and runoff estimation with respect to classical methods
based on the use of TWSA-only (suitable for river basins larger than 160'000 km², monthly time
scale) or on classical LSMs (Cai et al., 2014).
Moreover, although simple, the model has demonstrated a great potential to be easily applied over
sub-basins with different climatic and topographic characteristics, suggesting also the possibility to
extend its application to other basins. In particular, the analysis over basins with high human impact,
where the knowledge of the hydrological cycle and the river discharge monitoring is very important,
deserves special attention. Indeed, as the STREAM model is directly ingesting observations of soil
moisture and total water storage data, it allows the modeller to neglect processes that are implicitly
accounted for in the input data. Therefore, human-driven processes (e.g., irrigation, land use change),
that are typically very difficult to model due to missing information and might have a large impact
on the hydrological cycle, hence on runoff, could be implicitly modelled. The application of the
STREAM model on a larger number of basins with different climatic- physiographic characteristics
(e.g., including more arid basins, snow-dominated, lots of topography, heavily managed) along with
the results about the sensitivity analysis of the model parameters, will allow to investigate the
possibility to regionalize the model parameters and overcome the limitations of the automatic
calibration procedure highlighted in the discussion section.
**AUTHOR CONTRIBUTION**
S.C. performed the analysis and wrote the manuscript. G.G. collected the data and helped in
performing the analysis; C.M, L.B., A.T., N.S., H.H.F., C.M., M.R. and J.B. contributed to the
supervision of the work. All authors discussed the results and contributed to the final manuscript.

## CODE AVAILABILITY

The STREAM model version 1.3, with a short user manual, is freely downloadable in Zenodo (https://zenodo.org/record/4744984, doi: 10.5281/zenodo.4744984). The STREAM model code is distributed through M language files, but it could be run with different interpreters of M language, like the GNU Octave (freely downloadable here https://www.gnu.org/software/octave/download).

## DATA AVAILABILITY

All data and codes used in the study are freely available online. Air temperature data are available at https://psl.noaa.gov/data/gridded/data.cpc.globaltemp.html (last access 25/11/202). In situ river discharge data have been taken from the Global Runoff Data Center (GRDC, https://www.bafg.de/GRDC/EN/Home/homepage_node.html (last access 25/11/202). Precipitation and soil moisture data are available from http://pmm.nasa.gov/data-access/downloads/trmm and https://esa-soilmoisture-cci.org/, respectively.

## COMPETING INTERESTS

The authors declare that they have no conflict of interest.

## ACKNOWLEDGMENTS

The authors wish to thank the Global Runoff Data Centre (GRDC) for providing most of the streamflow data throughout Europe. The authors gratefully acknowledge support from ESA through the STREAM Project (EO Science for Society element Permanent Open Call contract n° 4000126745/19/I-NB).

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

Table 1. Location of river discharge gauging stations over the Mississippi basins and upstream
contributing area. Bold text is used to indicate gages where the STREAM model has been calibrated.

| # | River | Gage name | Latitude (°) | Longitude (°) | Upstream area (km²) | Mean annual river discharge (m³/s) | Presence of dam |
|---|---|---|---|---|---|---|---|
| 1 | Missouri | Bismarck, ND | -100.82 | 46.81 | 481232 | 633 | Garrison dam |
| 2 | Missouri | Omaha, NE | -95.92 | 41.26 | 814371 | 914 | Gavins Point Dam |
| 3 | Missouri | Kansas City, MO | -94.59 | 39.11 | 1229427 | 1499 | --- |
| 4 | **Missouri** | **Hermann, MO** | **-91.44** | **38.71** | **1330000** | **2326** | **---** |
| 5 | Kansas | Wamego, KS | -96.30 | 39.20 | 143054 | 141 | Kanopolis |
| 6 | **Mississippi** | **Keokuk, IA** | **-91.37** | **40.39** | **282559** | **1948** | **---** |
| 7 | Rock | Near Joslin, IL | -90.18 | 41.56 | 23835 | 199 | --- |
| 8 | Mississippi | Chester, IL | -89.84 | 37.90 | 1776221 | 6018 | --- |
| 9 | **Arkansas** | **Murray Dam Near Little Rock, AR** | **-92.36** | **34.79** | **408068** | **1249** | **---** |
| 10 | **Mississippi** | **Vicksburg, MS** | **-90.91** | **32.32** | **2866590** | **17487** | **---** |
| 11 | **Ohio** | **Metropolis, ILL.** | **-88.74** | **37.15** | **496134** | **7931** | **---** |


Table 2. Performance scores obtained over the Mississippi river gauging stations during the
calibration and validation periods.

| # | CALIBRATION PERIOD | | | VALIDATION PERIOD | | |
|---|---|---|---|---|---|---|
| **SCORE** | *KGE* (-) | rho (-) | *RRMSE* (%) | *KGE* (-) | rho (-) | *RRMSE* (%) |
| **GAUGING STATIONS USED FOR CALIBRATION** | | | | | | |
| **10** | 0.78 | 0.78 | 30 | 0.71 | 0.80 | 40 |
| **9** | 0.79 | 0.80 | 66 | 0.21 | 0.90 | 112 |
| **6** | 0.80 | 0.80 | 42 | 0.74 | 0.81 | 48 |
| **4** | 0.78 | 0.78 | 45 | 0.73 | 0.76 | 49 |
| **11** | 0.80 | 0.81 | 45 | 0.72 | 0.85 | 51 |
| **GAUGING STATIONS NOT USED FOR CALIBRATION** | | | | | | |
| **1** | -3.07 | 0.09 | 131 | 0.43 | 0.45 | 93 |
| **2** | -0.46 | 0.50 | 110 | 0.44 | 0.54 | 86 |
| **3** | 0.23 | 0.73 | 78 | 0.42 | 0.72 | 69 |
| **5** | -1.43 | 0.24 | 361 | -1.23 | 0.31 | 355 |
| **7** | 0.55 | 0.62 | 72 | 0.34 | 0.64 | 76 |
| **8** | 0.81 | 0.84 | 35 | 0.78 | 0.83 | 39 |


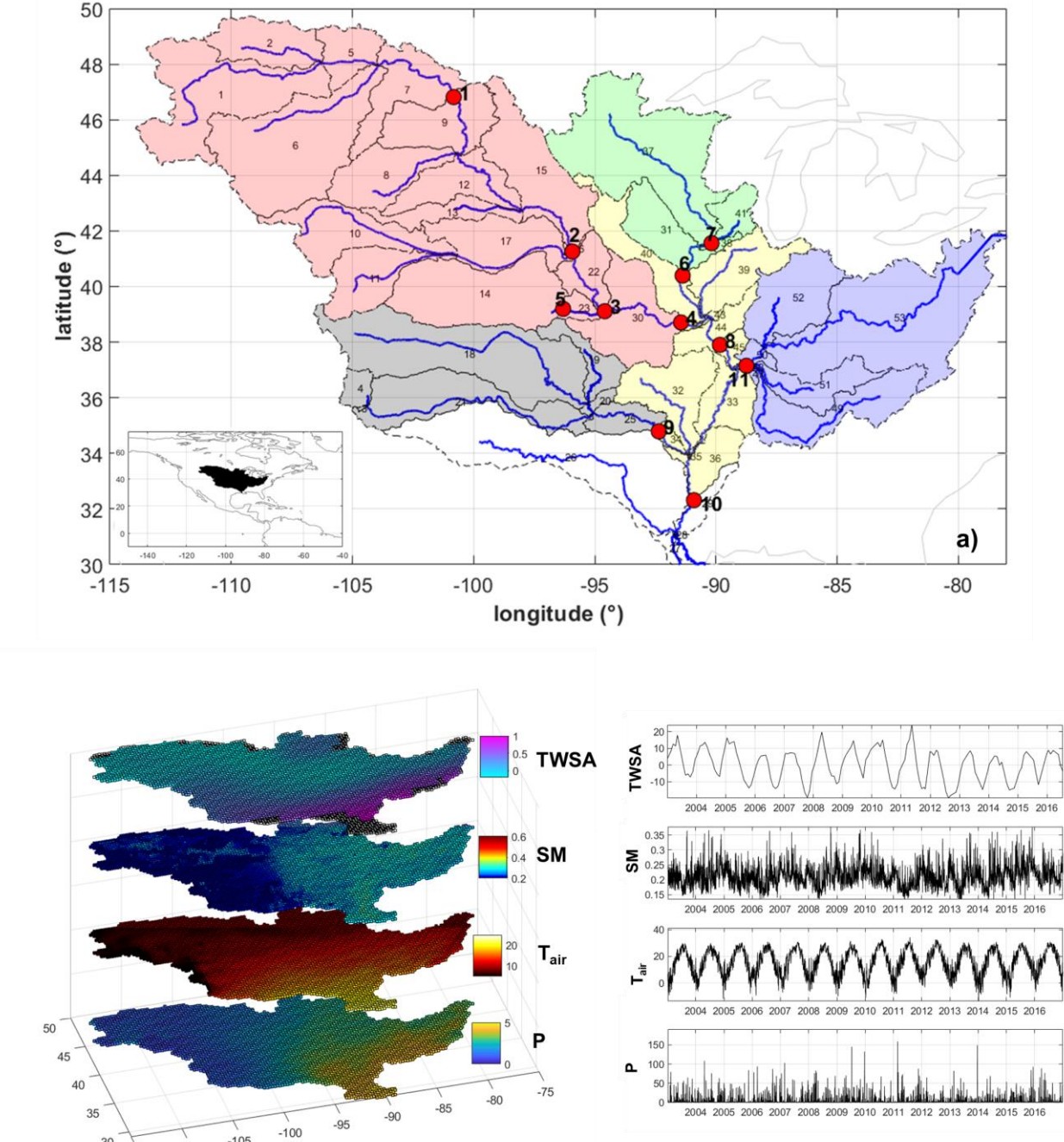

Figure 1. Mississippi river basin. Figure 1a) illustrates the sub-catchments delineation. The black dashed lines and the numbers in the map identify the 53 sub-catchments (tributary and directly draining areas) in the Mississippi basin, blue lines represent the mainstem of each sub-catchment. Red dots indicate the location of the river discharge gauging stations; different colours identify different inner cross-sections (and the related contributing sub- catchments) used for the model calibration. Figure 1b) shows the gridded mean daily values of the input data for the period 2003-2016. Figure 1c) illustrates the input time series over a point located inside the basin.

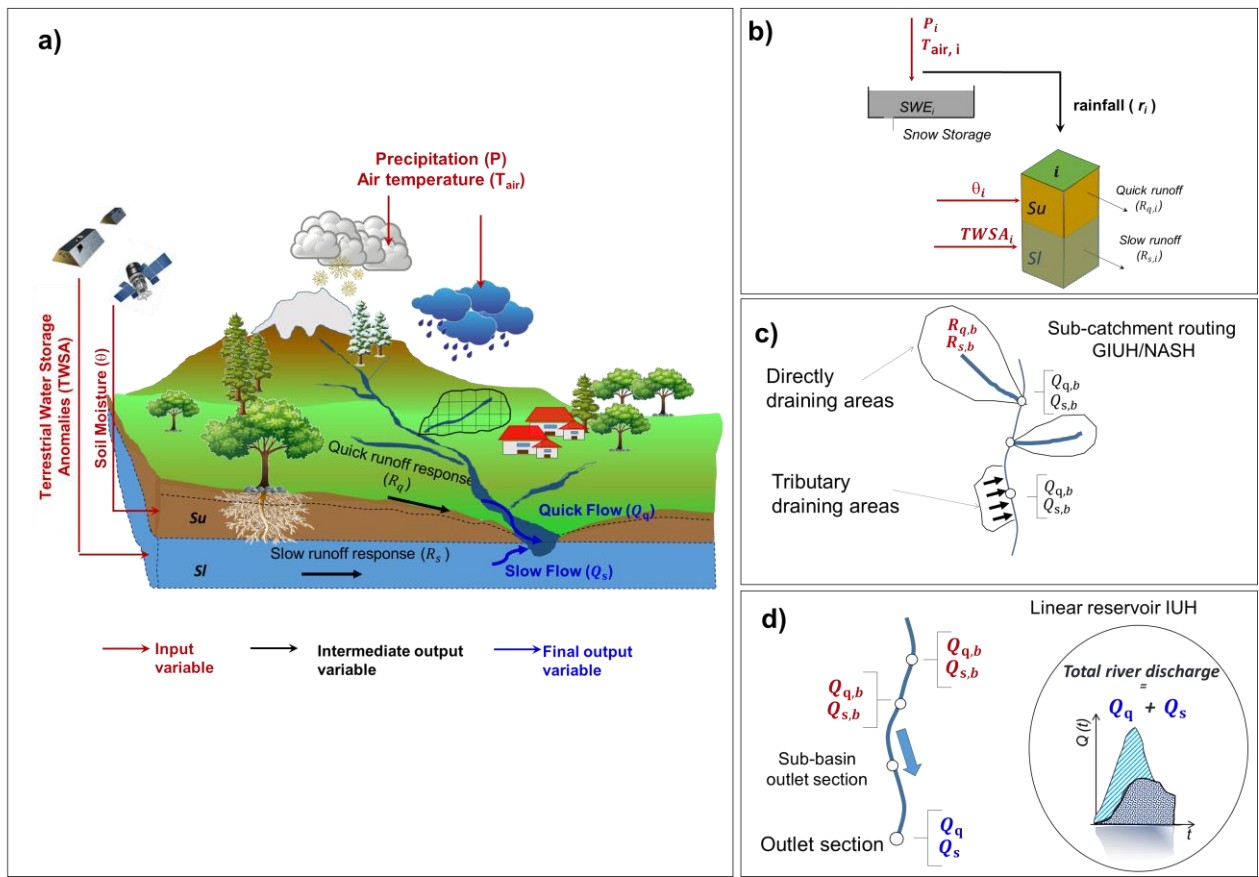

Figure 2. Configuration of the STREAM model adopted for runoff and river discharge estimation.
Figure 2a) gives an overview of the needed input data and the variables can be obtained as model
output. Figure 2b) illustrates the runoff generation at cell scale. Figure 2c) refers to the sub-catchment
river discharge calculation and Figure 2d) illustrates the river discharge routing through river
networks. Red arrows indicate input variables; black arrows indicate intermediate output variables;
blue arrows indicate final output variables. Please refer to text for symbols.


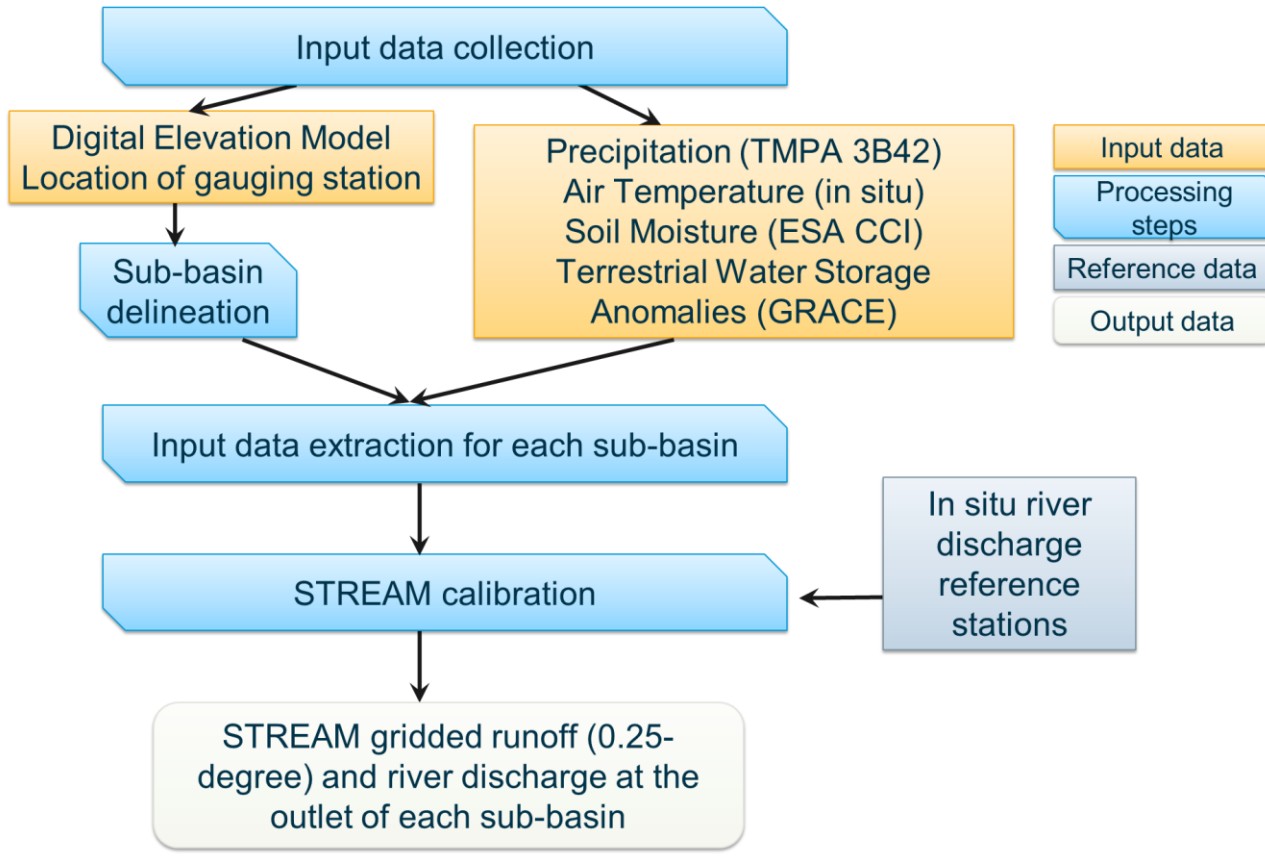

Figure 3. Processing steps of the STREAM model.

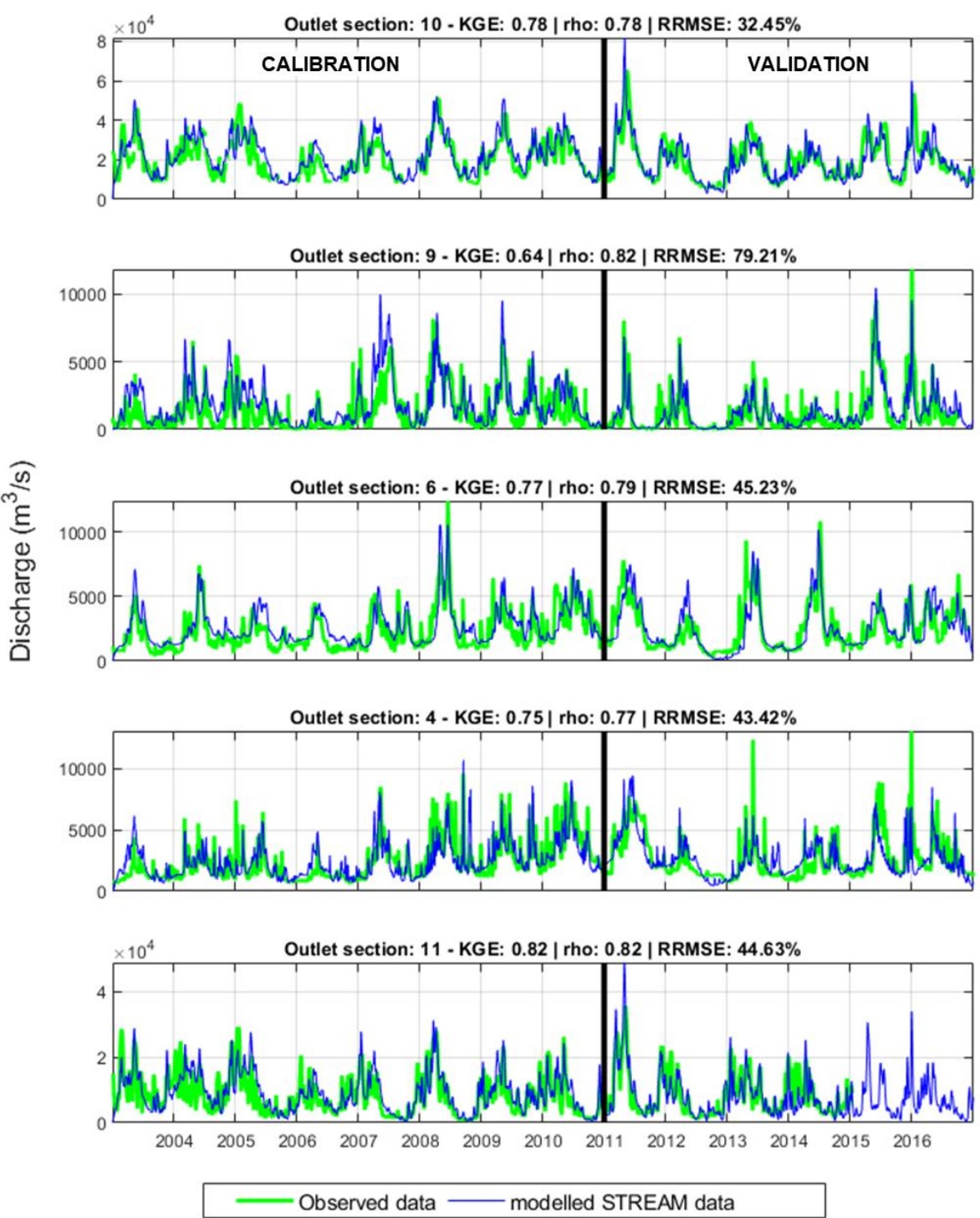

Figure 4. Comparison between observed and modelled river discharge time series over the five calibrated sections in the Mississippi river basin. Performance scores at the top of each plot refer to the entire study period (2003–2016).

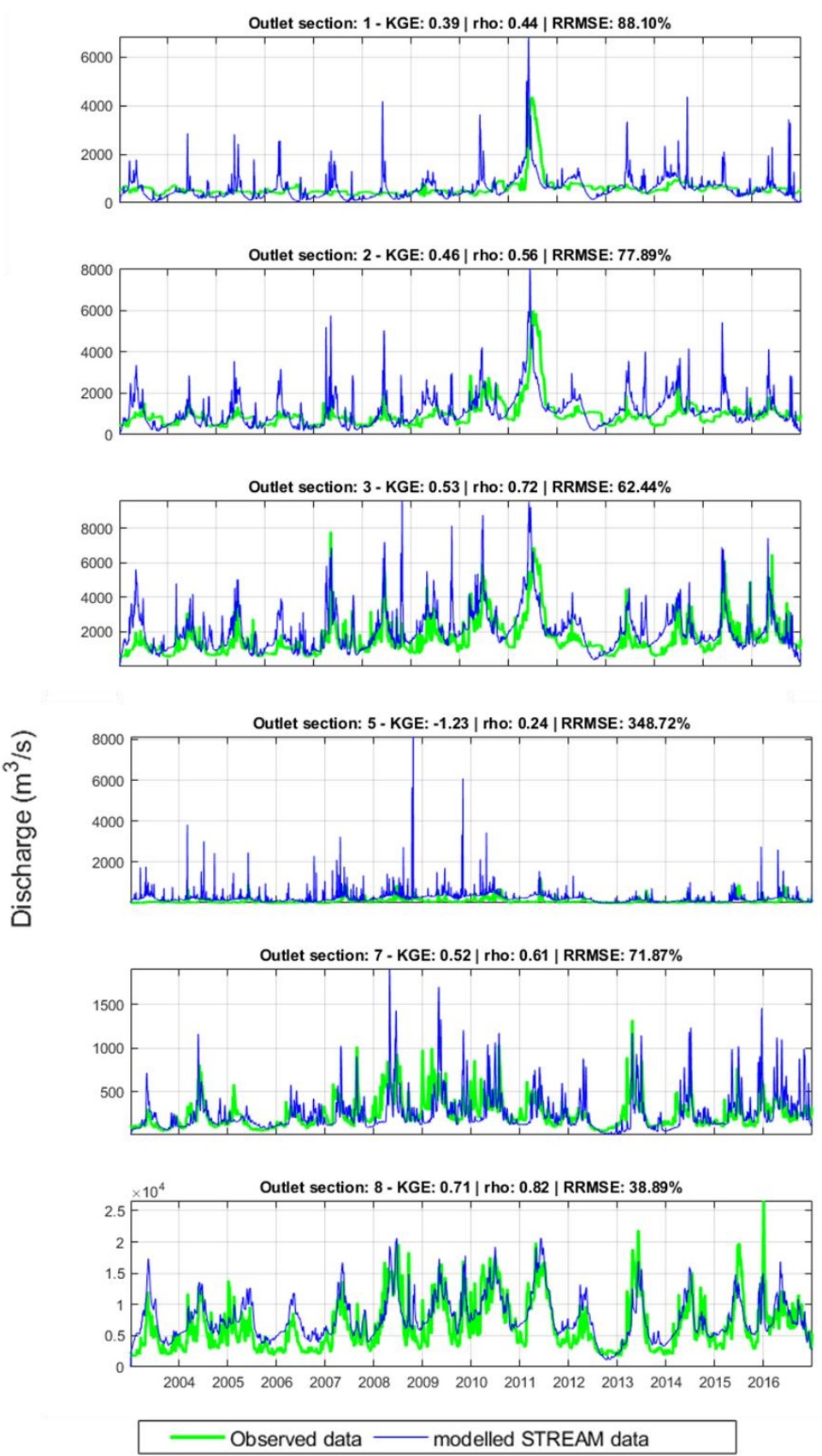

Figure 5. Comparison between observed and modelled river discharge time series over the gauged sections not used in the calibration phase. Performance scores at the top of each plot refer to the entire study period (2003–2016).

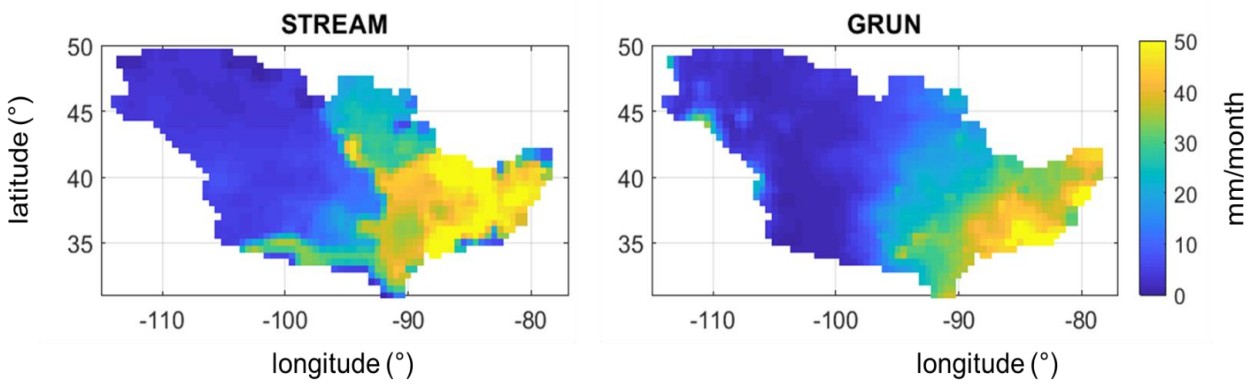


Figure 6. Mississippi river basin: mean monthly runoff for the period 2003−2014 obtained by
STREAM and GRUN models.

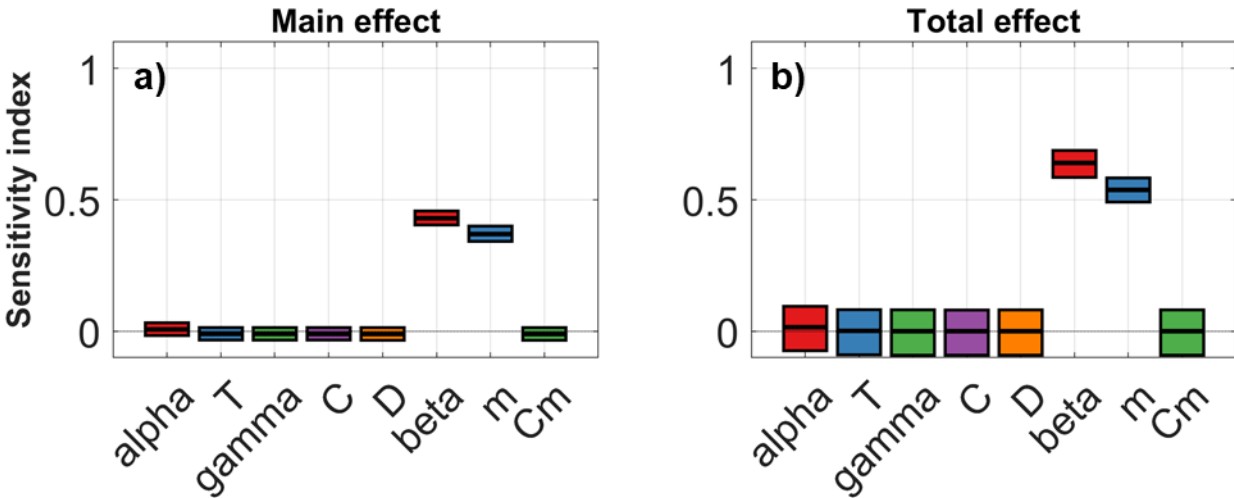

Figure 7. Main effect a) and total effect b) sensitivity indices calculated using the VBSA method for
Vicksburg gauging station. The boxes represent the 95% bootstrap confidence intervals and the
central black lines indicate the bootstrap mean.

**APPENDIX**
Table 1A. Description of STREAM parameters, belonging module, variability range and unit.

| Parameter | Description | Module | Range Variability | Unit |
|---|---|---|---|---|
| Cm | degree-day coefficient | Snow | 0.1/24-3 | [-] |
| α | exponent of infiltration | Soil | 1-30 | [-] |
| T | characteristic time length | Soil | 0.01-80 | [days] |
| β | coefficient relationship *slow-flow* runoff component and TWSA | Soil | 0.1-20 | [mm h$^{-1}$] |
| m | exponent in the relationship between *slow-flow* runoff component and TWSA | Soil | 1-15 | [-] |
| γ | parameter of GIUH | Routing | 0.5-5.5 | [-] |
| C | Celerity | Routing | 1-60 | [km h$^{-1}$] |
| D | Diffusivity | Routing | 1-30 | [km$^2$ h$^{-1}$] |



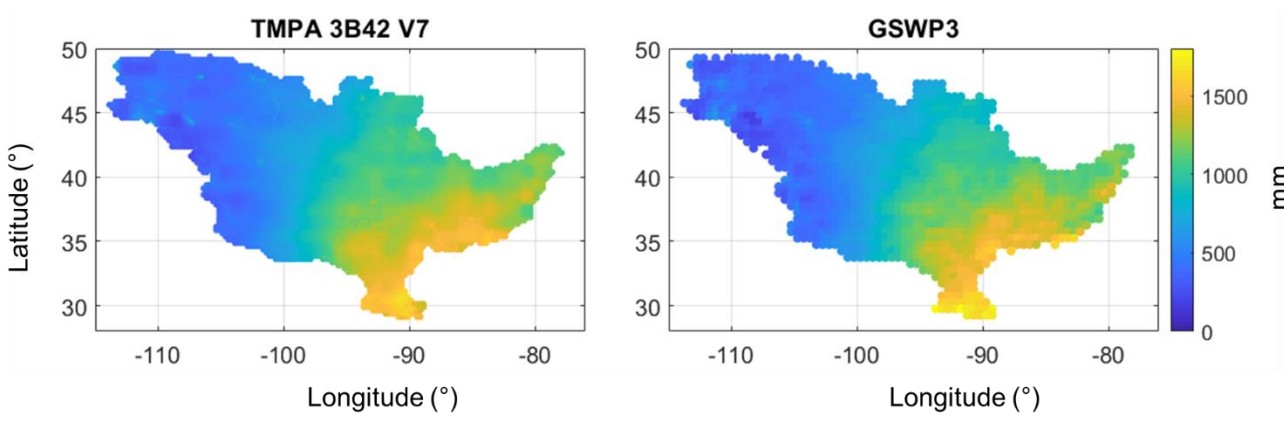


Figure A1. Mean annual precipitation data over the period 2003-2014 obtained by TMPA 3B42 V7
and GSWP3 datasets over the Mississippi river basin.