# Peer review of "SYNERGY BETWEEN SATELLITE OBSERVATIONS OF SOIL MOISTURE"

_Geoscientific Model Development, 2020_

## Referee Comment (RC2)

This paper presents a simple data-driven model, call STREAM, to estimate global runoff using satellite observations of precipitation, soil moisture, and total water storage anomalies (TWSA). The structure of the model is simple but clear — precipitation and soil moisture are used to estimate surface quick flow while TWSA is used to compute underground slow flow. It is shown that the model can be used to estimate runoff at a basin scale after careful calibrations, which is evidenced by the validation over five calibrated sections in Figure 4. However, I doubt very much whether the model can be used for the global runoff estimation since 8 parameters need to be well calibrated based on observed river discharge, which will, to a large extent, limit its application on a global scale. For example, the validation results over the gauged sections not used in the calibration phase do not show very good performance of the model as the difference between simulations and observed river discharge may go beyond 1000 m3/s in most sections. The authors attribute this difference to the presence of dams, but this may also happen in sections without dams such as sections 3 and 7.

On the other hand, this paper highlights the use of three satellite observations of precipitation, soil moisture, and TWSA. However, these three components are highly correlated with each other. For example, soil moisture can be used to estimate rainfall through the SM2RAIN algorithm [1]. Another example is that, on a regional scale, TWSA is very synchronous with soil moisture [2]. Accordingly, the synergy between precipitation, soil moisture, and TWSA, to me, shall be very limited. For these reasons, I suggest rejecting this paper as is.

[1] Luca Brocca et al (2015). Rainfall estimation from in situ soil moisture observations at several sites in Europe: an evaluation of the SM2RAIN algorithm. HESS.
[2] A Geruo et al (2017). Satellite-observed changes in vegetation sensitivities to surface soil moisture and total water storage variations since the 2011 Texas drought. ERL.

Minor comments:

1. As the experiments are only conducted over the Mississippi river basin, the word "GLOBAL" used in the title may not be suitable.

2. In line 122, please add necessary references regarding SMAP and GPM.

3. Please add some necessary references to Eqs. 1 and 4.

4. The statements in lines 295-296 are slightly in conflict with the statements in lines 306-308. As I know, TWSA can partly include information on soil moisture.

5. In line 303, what are the ranges of beta and m values?

6. In line 344, the meaning of the Horton-Strahler order is not clear.

7. In lines 444-445, the authors mention that the performance of model in section 3 is not bad. However, as I checked from Figure 5, the difference between simulated and observed discharge can go beyond 8000 m3/s.

---

## Author Response (AR1)

**Review of "Synergy between satellite observations of soil moisture and water storage anomalies for global runoff estimation"**

We thank the reviewers and the Editor for their supportive review. In the revised version of the manuscript the following relevant changes have been made:

- the title has been changed in "Synergy between satellite observations of soil moisture and water storage anomalies for runoff estimation" (see comment#2 of RC2 or comment#3 of RC3).
- To avoid any misunderstanding, the STREAM model has been defined as a conceptual hydrological model instead of a data-driven model (see comment#4 of RC2).
- The section "code availability" has been modified specifying that "STREAM model code version 1.3 has been stored in Zenodo (https://zenodo.org/record/4744984, doi: 10.5281/zenodo.4744984)". The authors have included the name of the model and the version throughout the revised manuscript (see EC comments).
- In Zenodo the authors provided a tutorial (a markdown "README.rst" file), describing the input and output files as well as the codes to be run for the simulation of river discharge and runoff. Moreover, in the "README.rst" file the authors specified that STREAM codes are distributed through M language files that can be run through different interpreters such as the free Octave (I tried to run the code by using Octave and it works).
- References and details on the equations have been added according to the reviewer's suggestions (see comment#8 of RC2).
- A Table (Table 1A) has been added in appendix to better describe the STREAM parameters and their range of variability (see comment#10 of RC2).

**EC: Comments and replies**

1. CEC1: Dear authors,

   You can not publish the code of the model after the acceptance of the paper. It must be available during the review process. Therefore, please, make it available in a permanent repository—for example, Zenodo. As soon as you have done it, post a comment to your manuscript, including the model version number and the link to the Zenodo repository and corresponding DOI.

   Also, please, in the next step of the review process, be sure that you include the name of the model and the version that you use in the title of the manuscript.

   Best regards,

   Juan A. Añel

   AC: The STREAM model code version 1.3 has been stored in Zenodo (https://zenodo.org/record/4744984, doi: 10.5281/zenodo.4744984). In particular, following your suggestion, in the "README.rst" file we specified that STREAM codes are distributed through M language files that can be run through different interpreters such as the free Octave (I tried to run the code by using Octave and it works). Details are given in the revised manuscript at the section "Code availability".

2. EC2: For purposes of this paper, I would recommend that you use the DOI of the version of record. This is in order to ensure that the model described in this manuscript matches the source code and repository, and that future changes may be made without breaking that link.

   AC: The authors thank the editor for the suggestion. In the revised version of the paper, the DOI of the version of record has been used. Details are given in the section "Code availability" of the manuscript as follows:

   "The STREAM model version 1.3 , with a short user manual, is freely downloadable in Zenodo (https://zenodo.org/record/4744984, doi: 10.5281/zenodo.4744984)."

3. EC2: In addition, GMD requires a user's manual as part of the manuscript. I see here (\url{https://github.com/IRPIhydrology/STREAM}) that you have a small ``tutorial''. However, while this describes variables (quite useful), it does not actually give instructions for how to run the model, which is more of what I would expect when I see a ``tutorial''. I would recommend that you move all of this material to the README, and I ask that you generate a more full-featured user's guide as documentation.

   AC: In agreement with the Editor suggestion, on Github (https://github.com/IRPIhydrology/STREAM) and on Zenodo (https://zenodo.org/record/4744984, doi: 10.5281/zenodo.4744984), the authors have provided an updated document "README.rst", describing the input and output files as well as the codes to be run for the simulation of river discharge and runoff.

4. EC2: Thank you for the updated tutorial. For the purposes of presentation -- though beyond the scope of what I request as an editor -- you may want to consider using the README markdown file instead of your own txt file. This allows for better formatting and increased readability.

AC: Following your suggestion, on Github (https://github.com/IRPIhydrology/STREAM) and on Zenodo (https://zenodo.org/record/4744984, doi: 10.5281/zenodo.4744984) a markdown file "README.rst" has been added.

**RC1: Comments and replies**

1. R1: The paper is very thorough in presenting the novel and useful STREAM model. The paper is well-organized, clearly presented, and contains all necessary information for the reader to conceptualize and understand the design and validation. I am very interested in any future work applying this model to other global basins and its performance with respect to land surface models that require substantially more parameterizations and computational load.

   AC: The authors are thankful to the reviewer for their assessment of our paper. Really appreciated!

**RC2: Comments and replies**

1.  RC2: This paper presents a simple data-driven model, call STREAM, to estimate global runoff using satellite observations of precipitation, soil moisture, and total water storage anomalies (TWSA). The structure of the model is simple but clear — precipitation and soil moisture are used to estimate surface quick flow while TWSA is used to compute underground slow flow. It is shown that the model can be used to estimate runoff at a basin scale after careful calibrations, which is evidenced by the validation over five calibrated sections in Figure 4.

    AC: The authors are thankful to the reviewer for their assessment of our paper. We have provided a point-by-point reply to each of the comments in the sequel.

2.  RC2: However, I doubt very much whether the model can be used for the global runoff estimation since 8 parameters need to be well calibrated based on observed river discharge, which will, to a large extent, limit its application on a global scale.

    AC: The intent of the paper is to describe a model that could be used for the estimation of river flow (and runoff) worldwide. However, as correctly stated by the reviewer, the model results are only shown for the Mississippi River basin and the "Global" in the title may not be appropriate. Consistent with the reviewer's assertion, the article and model should be evaluated for their ability to estimate river flow and runoff in the Mississippi River Basin, not globally. In the revised version of the manuscript, the term "Global" from the title has been removed. The new title is: "Synergy between satellite observations of soil moisture and water storage anomalies for runoff estimation".

    The possibility to regionalize the model parameters for the estimation of river discharge at global scale is a topic beyond the scope of this article. However, the authors are working on the regionalization of the model parameters. Preliminary results, which have been shown at the EGU 2021 conference, demonstrate the possibility to link model parameter values to the basin characteristics, still obtaining satisfactory results. Future studies will address the problem of the regionalization of the model parameters. This aspect has been specified in the revised version of the manuscript (see Lines 473-474):

    *"A more in-depth investigation about the model calibration procedure and the regionalization of the model parameters will be carried out in future studies."*

3.  RC2: For example, the validation results over the gauged sections not used in the calibration phase do not show very good performance of the model as the difference between simulations and observed river discharge may go beyond 1000 m3/s in most sections. The authors attribute this difference to the presence of dams, but this may also happen in sections without dams such as sections 3 and 7.

    AC: As any hydrological model calibrated against observed data, it is expected that the best performances of STREAM model can be obtained over the calibrated gauging sections whereas the performances will decrease over the gauging sections not used for the calibration. However, gauging section 3 and section 7 cannot be taken as reference to understand if the STREAM model is suitable for reproducing river discharge over not calibrated sections as they are affected by local characteristics (section 7 is located over the Rock river (near Joslin), a relatively small tributary of Mississippi river, 23'000 km$^2$, if compared with GRACE resolution, 160'000 km$^2$) and by the presence of an important dam (section 3).

The manuscript has been modified accordingly (see Lines 460-465):

*"In particular, over sections 3 (influenced by the presence of dams in section 1 and 2) and 7 (located over the Rock river a relatively small tributary of Mississippi river see Table 1), the STREAM model overestimates the observed river discharge highlighting that the model parameters estimated for river section 4 and 6, respectively, are not suitable to accurately reproduce river discharge for river sections 3 and 7 (see Figure 3 and Figure 5)."*

*and (see Lines 469-473 ):*

*"Although it is expected that the performances of STREAM model, as any hydrological model calibrated against observed data, can decrease over the gauging sections not used for the calibration, the findings obtained above, raises doubts about the robustness of model parameters and whether it is actually possible to transfer model parameters from one river section to another with different interbasin characteristics. A more in-depth investigation about the model calibration procedure and the regionalization of the model parameters will be carried out in future studies."*

Moreover, we underline that a decrease in the model performance is obtained for all hydrological models. We are preparing a second paper showing the comparison between STREAM and other global hydrological model performances, and STREAM model is working similarly (and even better) than other models (much more complex and with a much larger number of parameters). Preliminary results have been shown at EGU conference in 2020 (https://doi.org/10.5194/egusphere-egu2020-13718).

4.  RC2: On the other hand, this paper highlights the use of three satellite observations of precipitation, soil moisture, and TWSA. However, these three components are highly correlated with each other. For example, soil moisture can be used to estimate rainfall through the SM2RAIN algorithm [1]. Another example is that, on a regional scale, TWSA is very synchronous with soil moisture [2]. Accordingly, the synergy between precipitation, soil moisture, and TWSA, to me, shall be very limited.

    AC: The reviewer is right that precipitation, soil moisture and TWSA are in some way correlated but this does not represent a problem for our approach. The proposed STREAM model is a conceptual hydrological model where the inputs contribute to the different runoff components according to specific laws. Specifically, soil moisture and precipitation contribute to the quick component of runoff (daily time scale) while TWSA contributes to the slow component (monthly scale). The differences in the temporal (and spatial) scale of the input data allow us to use the different input consistently and to optimize their synergy for runoff estimation.

    This aspect has been better specified in the text (see Lines 236-250):

    *"The concept behind the STREAM model is that river discharge is a combination of hydrological responses operating at diverse time scales (Blöschl et al., 2013; Rakovec et al., 2016). In particular, river discharge can be considered made up of a slow-flow component, produced as outflow of the groundwater storage and of a quick-flow component, i.e. mainly related to the surface and subsurface runoff components (Hu and Li, 2018). While the high spatial and temporal (i.e., intermittence) variability of precipitation and the highly changing land cover spatial distribution significantly impact*

*the variability of the quick-flow component (with scales ranging from hours to days and meters to kilometres depending on the basin size), slow-flow river discharge reacts to precipitation inputs more slowly (i.e., months) as water infiltrates, is stored, mixed and is eventually released in times spanning from weeks to months. Therefore, the two components can be estimated by relying upon two different approaches that involve different types of observations. Based on that, within the STREAM model, satellite soil moisture, precipitation and TWSA will be used for deriving river discharge and runoff estimates. The first two variables are used as proxy of the quick-flow river discharge component while TWSA is exploited for obtaining its complementary part, i.e., the slow-flow river discharge component."*

Likely, the misunderstanding could have been generated as in the text the STREAM model is defined as a "data-driven model" to indicate that the model is mainly based on the contribution of the input data rather than of complex equations and processes. To avoid any misunderstanding, in the revised version of the manuscript the model has been defined as conceptual hydrological model and "data-driven" has been removed.

5. RC2: For these reasons, I suggest rejecting this paper as is.

   AC: We hope that with these new explanations the reviewer might reconsider their decision.

   RC2: [1] Luca Brocca et al (2015). Rainfall estimation from in situ soil moisture observations at several sites in Europe: an evaluation of the SM2RAIN algorithm. HESS. [2] A Geruo et al (2017). Satellite-observed changes in vegetation sensitivities to surface soil moisture and total water storage variations since the 2011 Texas drought. ERL.

   Minor comments:

6. RC2: As the experiments are only conducted over the Mississippi river basin, the word "GLOBAL" used in the title may not be suitable.

   AC: The title of the revised manuscript has been modified as: "Synergy between satellite observations of soil moisture and water storage anomalies for runoff estimation".

7. RC2: In line 122, please add necessary references regarding SMAP and GPM.

   AC: The references for SMAP and GPM have been added to the revised manuscript.

8. RC2: Please add some necessary references to Eqs. 1 and 4.

   AC: The references for equations 1 and 4 have been added to the revised manuscript. In the manuscript you can read (see Lines 278-280):

   "Consequently, the quick runoff response, Qfu from the first storage can be computed through equation following the formulation proposed by Georgakakos and Baumer (1996), as in equation (1):"

   and Lines 299-301:

"For that, the slow runoff response Qsl, from the second storage, can be computed following the formulation proposed by Famiglietti and Wood (1994), through equation (4) as follows:"

9. RC2: The statements in lines 295-296 are slightly in conflict with the statements in lines 306-308. As I know, TWSA can partly include information on soil moisture.

The two statements have been slightly modified to underline that TWSA can partly include information on soil moisture. However, by considering the different time scales at which the quick and slow runoff responses act, in the STREAM model we are making the assumption that they are independent. This aspect has been specified in the revised version of the manuscript. In particular, the statement in Lines 295-295 (see Lines 298-299 in the new manuscript) has been modified as in the following:
"*Indeed, the time scale of slow runoff response is typically in the range of seasons to years and it can be assumed almost independent upon the water that is contained in that upper storage*".

The statement in Lines 306-308 has been modified as follows (see Lines 309-312 of the new manuscript):

"*Note that we made the hypothesis that soil moisture and TWSA observations are independent (whereas in the reality soil moisture can be responsible both for the generation of the quick flow part (mainly) and for the slow flow contribution) given the different temporal (and spatial) scales at which the quick and slow runoff responses act.*"

10. RC2: In line 303, what are the ranges of beta and m values?

AC: A Table (Table 1A) describing the STREAM parameters, the module to which they belong, the variability range and unit has been added in the Appendix of the revised manuscript.

11. RC2: In line 344, the meaning of the Horton-Strahler order is not clear.

AC: The sentence has been modified as (see Lines 349-350): "*by considering only rivers with order greater than 3 (according to the Horton-Strahler rules, Horton, 1945; Strahler, 1952)*" and references have been added to the revised manuscript.

12. RC2: In lines 444-445, the authors mention that the performance of model in section 3 is not bad. However, as I checked from Figure 5, the difference between simulated and observed discharge can go beyond 8000 m3/s.

AC: On overall, the performance of the model in section 3 of Figure 5 has been defined as "not bad" because the KGE value obtained for the 2003-2016 period, between observed and simulated river discharge, is equal to 0.48. However, by looking at the time series it can be noted that sometimes the STREAM model largely overestimates the observed river discharge. This issue could be due to:

1) observed river discharge influenced by the presence of upstream dams. Indeed, observed river discharge time series from section 1, 2 and 3 are quite similar each other and clearly influenced by the reservoir management (see e.g. flood peak on 2011 or the time series between 2012-2014); however the STREAM model does not include specific module for modelling reservoirs; 2) the model parameters are estimated for a different section, so the basin characteristics could have an impact on the model results.

This aspect has been specified in the revised manuscript (see Lines 459-464) as follows:

*"Positive KGE values are obtained over river sections 3, 7 and 8. In particular, over sections 3 (influenced by the presence of dams in section 1 and 2) and 7 (located over the Rock river a relatively small tributary of Mississippi river see Table 1), the STREAM model overestimates the observed river discharge highlighting that the model parameters estimated for river section 4 and 6, respectively, are not suitable to accurately reproduce river discharge for river sections 3 and 7 (see Figure 3 and Figure 5)."*

**RC3: Comments and replies**

1. RC3: The authors present a model 'STREAM' that is used to derive river discharge and runoff. The STREAM model is conceptually and computationally simple, and uses inputs of precipitation, total water storage, soil moisture as well as air temperature to provide estimates of global runoff. The results are tested for the Mississippi River basin in the United States and indicate good agreement. Enhancing the ability to model distributed runoff has important applications for hydrology. However, further justification of the methods used and applicability to other climate regimes and regions is needed.

   AC: The authors are thankful to the reviewer for their assessment of our paper. We have provided a point-by-point reply to each of the comments in the sequel. Additional details and justifications of the method used will be added in the revised version of the manuscript.

2. RC3: For example, the authors should comment on the sensitivity of the model to the hydrological inputs of precipitation, soil moisture and total water storage anomaly. The authors should comment on the contribution of using these inputs, and whether results are improved or not by using all three. Otherwise it is not immediately clear to the reader the contribution of each to the estimation of runoff.

   AC: In a preliminary analysis, we have tested the sensitivity of STREAM model to the different hydrological inputs of precipitation, soil moisture and total water storage anomaly (not shown for brevity). In particular, by running the STREAM model with different input configurations (e.g., by using TRMM3B42 or CPC data for precipitation, ESA CCI or ASCAT data for soil moisture, TWSA or soil moisture data to simulate the slow-flow river discharge component), we found that STREAM results are more sensitive to soil moisture data rather than to precipitation input. In addition, by running STREAM model with soil moisture data as input to simulate the slow-flow river discharge component (i.e. without using TWSA data) we found a deterioration of the model results.

   A sentence to explain the contribution of the hydrological inputs to the STREAM results has been added in the revised version of the manuscript (see Lines 498-511):

   *"This is an important result of the study as it demonstrates, on one hand, that the model structure is appropriate with respect to the data used as input and, on the other hand, the great value of information contained into TWSA data that, even if characterized by limited spatial/temporal resolution, can be used to simulate runoff and river discharge at basin scale. This finding has been also confirmed by a preliminary sensitivity analysis in which the STREAM v1.3 model has been run with different hydrological inputs of precipitation, soil moisture and total water storage anomaly (not shown here for brevity). In particular, by running the STREAM model with different input configurations (e.g., by using TMPA 3B42 V7 or Climate Prediction Center (CPC) data for precipitation, ESA CCI or Advanced SCATterometer (ASCAT) data for soil moisture, TWSA or soil moisture data to simulate the slow-flow river discharge component), we found that STREAM results are more sensitive to soil moisture data rather than to precipitation input. In addition, by running STREAM model with soil moisture data as input to simulate the slow-flow river discharge component (i.e. without using TWSA data) we found a deterioration of the model results."*

3. RC3: The authors also only test their model in the Mississippi River basin, however it would be interesting and informative to address the performance of this model in different regions including more arid basins, snow-dominated, lots of topography, heavily managed, etc. The study indicates it is a 'global' model so more discussion of its applicability worldwide is needed.

AC: The intent of the paper is to describe a model that could be used for the estimation of river flow (and runoff) worldwide. However, as correctly stated by the reviewer, the model results are only shown for the Mississippi River basin and even if the model could be applied at global scale, the "Global" in the title may not be appropriate for this manuscript. For that, in the revised version of the manuscript, "Global" in the title has been removed.

Concerning the applicability of the model to other climate regimes and regions, the authors are preparing a new manuscript where STREAM model has been tested on 5 pilot basins (Mississippi, Amazon, Niger, Danube and Murray-Darling) across the world with good model performances. Preliminary results have been shown at EGU conference in 2020 (https://meetingorganizer.copernicus.org/EGU2020/EGU2020-13718.html) and 2021 (https://meetingorganizer.copernicus.org/EGU21/EGU21-14175.html).

More specific comments on the applicability of the model in different regions including more arid basins, snow-dominated, lots of topography, heavily managed will be addressed in a future manuscript. This aspect has been specified in the revised version of the manuscript (see Lines 594-599):

*"The application of the STREAM v.1.3 model on a larger number of basins with different climatic- physiographic characteristics (e.g., including more arid basins, snow-dominated, lots of topography, heavily managed) will be object of future studies and it will allow to investigate the possibility to regionalize the model parameters and overcome the limitations of the automatic calibration procedure highlighted in the discussion section."*
* * *
Specific comments:

4. RC3: Line 208 - Can the authors provide the depth of the soil moisture used in this study.

AC: Satellite soil moisture observations obtained by the ESA CCI soil moisture product refer to first centimeters of soil (2-3 cm). However, in the STREAM formulation we used a root-zone soil moisture product derived by the surface ESA CCI satellite soil moisture product, by applying the exponential filtering approach in its recursive formulation as in Albergel et al. (2009). For that, the root zone soil moisture used in the STREAM model is referred to the first layer of the model, whose depth varies approximatively from 5 to 30 cm.

This aspect has been specified in the manuscript (see Lines 205-208):

*"Soil moisture data have been taken from the European Space Agency Climate Change Initiative (ESA CCI) Soil Moisture project (https://esa-soilmoisture-cci.org/) that provides a surface soil moisture product (referred to first 2-3 centimeters of soil) continuously*

*updated in term of spatial-temporal coverage, sensors and retrieval algorithms (Dorigo et al., 2017)."*

*and Lines (283-286):*

*"SWI is the Soil Water Index (Wagner et al., 1999), i.e., the root-zone soil moisture product referred to the first layer of the model (representative of the first 5-30 centimeters of soil), derived by the surface satellite soil moisture product, $\theta$, by applying the exponential filtering approach in its recursive formulation (Albergel et al., 2009):"*

5.  RC3: Line 262 - Given that the authors only validate in the Mississippi basin, can they comment on how different climate regimes could impact the accuracy of the modeled runoff in particular more snow-dominated basins. Can the authors comment on the validation of the snow module.

    AC: The Mississippi basin contains different climatic regimes, particularly if we consider the differences between the eastern and western parts of the basin. The authors are preparing a new manuscript where STREAM model has been tested on 4 additional basins (Amazon, Niger, Danube and Murray-Darling) thus exploring a larger variability of climatic regimes, as well as soil and land use characteristics.

    The impact of snow on runoff is difficult to evaluate at the scales of the grid considered in the study (25 km). Mountainous areas which are mainly located in the north western and eastern part of the Mississippi basin (i.e., the Rocky Mountains and Appalachian Mountains) receive significant precipitation as snow during the winter period and might provide a significant portion of snowmelt runoff during the warm season. While a comparison with point stations would be not meaningful at the working scale, the river discharge simulations at section11 do show relatively good agreement with observations during both calibration and validation periods suggesting the ability of the proposed approach to simulate snowmelt contribution.

6.  RC3: Line 300 - Can the authors provide additional information on how they resolve the differences in spatial and temporal scale between the various input data sets provided. In particular, the coarser scale of the GRACE data.

    AC: Concerning the spatial scale, air temperature, soil moisture, precipitation and GRACE data have been resampled over the precipitation spatial grid at 0.25° resolution through a bilinear interpolation. For the temporal scale, air temperature soil moisture and precipitation data are available at daily time step, while monthly GRACE data have been linearly interpolated at daily time step. Major details on how the differences in spatial and temporal scale between the various input data sets is resolved will be added in the revised version of the manuscript (see Lines 360-367).

    *"If characterized by different spatial/temporal resolution, these datasets need to be resampled over a common spatial grid/temporal time step prior to be used as input into the model. To run the STREAM v1.3 model over the Mississippi river basin, input data have been resampled over the precipitation spatial grid at 0.25° resolution through a bilinear interpolation. Concerning the temporal scale, $T_{air}$, soil moisture and precipitation data are available at daily time step, while monthly TWSA data have been linearly interpolated at daily time step. For each of the 53 Mississippi subbasins, the resampled precipitation, soil moisture, $T_{air}$ and TWSA data have been extracted."*

7.  RC3: Line 462 - can the model be run with same input precipitation as GRUN for the validation purposes? Or can the authors comment on precipitation differences between either product .

AC: The STREAM model could be run with the same input precipitation as GRUN to compare the runoff maps obtained by the two approaches. However, as it is beyond the purpose of the paper, the authors have added only a comment in the revised version of the manuscript to underline that the precipitation estimates used in GRUN and STREAM are very similar in terms of spatial pattern and mean annual rainfall amount. In particular, both the Global Soil Wetness Project Phase 3 (GSWP3) and the TRMM 3B 42 precipitation products clearly identify two distinct zones in the Mississippi basin, a western dry and an eastern wet area.

This aspect has been specified in the revised version of the manuscript (see Lines 479-485):

"*Although the two datasets consider different precipitation inputs, the two models agree in identifying two distinct zones in terms of runoff, i.e., the western dry and the eastern wet area. This two distinct zones can be clearly identified also in the GSWP3 and TMPA 3B42 V7 precipitation maps (not shown here) used as input in GRUN and STREAM, respectively, stressing that STREAM runoff output is correctly driven by the input data. However, likely due to the calibration procedure, the STREAM runoff map appears patchier with respect to GRUN and discontinuities along the sub-basin boundaries (identified in Figure 3) can be noted.*"

8.  RC3: Line 520 - Does using GRACE data for water storage (which captures both human and natural processes) address this? GRACE can indicate human activity and water extraction practices, which I think could help improve purely 'natural' estimates of runoff.

AC: The added value of satellite observations in the STREAM model is the possibility to capture processes and human activities not modelled but directly observed by satellite. However, we think that GRACE could not be appropriate for capturing reservoir management and for that, in the current development of STREAM model, we are including additional modules for simulating the presence of reservoirs and diversions along the river that can be relevant in several basins/regions.

---

## Referee Report (RR1)

In this study, Camici and coauthors present a simplified conceptual discharge model that uses precipitation, soil moisture, and temperature to model quick runoff and GRACE-derived storage changes to model slow runoff. Although the rational behind the work is solid, though not particularly novel (https://www.sciencedirect.com/science/article/pii/S0012821X08006766, https://link.springer.com/chapter/10.1007/978-3-030-02197-9_1), the results are only good in the basin it's calibrated over, with little potential for transfer. The coauthors attempt to validate their model's utility by expressing it's easy of use, computational efficiency, and limited input data requirements, but it is far from the only model to check these boxes. Without comparison to some more commonly used models, say VIC, SWAT, Sacramento, or HEC-HMS, it's hard to convince people that they should use the presented STREAM model. I strongly encourage the coauthors to compare their results with other simplified conceptual discharge models to validate their model's utility.

Line 278: The rain/snow diffentiation model should be expanded on within the study. Rain/snow differeation based on temperature and elevation is passably good, but at a large grid size like 25 x 25 km, the topographic complexity of higher elevations is lost. A differentiation scheme like that used in IMERG may be preferred, but isn't necessary. Still, this should be acknowledged, however briefly.

345-348: Using a calibration tool would be preferable to manually adjusting to maximize Kling-Gupta. Perhaps one was used, but its not specified. Also, does paragraph 5.1 relate to calibration, or is it paragraph 5.4?

Section 5.1: "1. Input data collection" is unnecessary to include.

Line 414-415: It is not clear to me what "to get to the right answers for the right reasons" means in this context and its tedious to hunt it down in the cited paper.

Line 500-501: I would encourage you to include a preciptiation map as a figure to illustrate your point.

Line 595: By the author's own admission (Lines 486-490), the model may not be suitible to reproduce discharge in basins not calibrated over. This should be changed to something less absolute. "Under some circumstances, the STREAM model can be used to estimate discharge in basins not calibrated over, especially those without upstream dams with comparable size and land cover." Or something similar.

---

## Referee Report (RR2)

[referee-annotated manuscript omitted]

---

## Author Response (AR2)

**Review of "Synergy between satellite observations of soil moisture and water storage anomalies for global runoff estimation"**

We thank the Topical Editor (TE) for their supportive review. In the following, the author replies (AC, blue lines) to the TE comments (red lines) are reported.

**TEC: Comments and replies**

Dear Dr. Camici and co-authors,

Thank you for your response to the referee comments. I am sending it out for re-review, and am also attaching the following comments of my own, which I would like you to address alongside any new referee comments.

TE1: Please remove all references to future work. We cannot know the future. I think that you can communicate the same and/or similar points about applicability without invoking your specific future plans.

AC: Any references to future works have been removed in the revised manuscript. Specifically, Lines 472-473 of the manuscript:

"A more in-depth investigation about the model calibration procedure and the regionalization of the model parameters will be carried out in future studies."

have been modified as (see Lines 490-492 of the revised manuscript):

"A more in-depth investigation about the model calibration procedure, with special focus on the regionalization of the model parameters, should be carried out but this topic is beyond the scope of the manuscript."

Line 569:

"However, this aspect is beyond the paper purpose and it will conveniently be addressed in future works."

has been modified as (see Lines 587-589 of the revised manuscript):

"As this aspect requires additional investigations and it is beyond the paper purpose, it will not be tackled here."

Lines 594-599:

"The application of the STREAM v1.3 model on a larger number of basins with different climatic- physiographic characteristics (e.g., including more arid basins, snow-dominated, lots of topography, heavily managed) will be object of future studies and it will allow to investigate the possibility to regionalize the model parameters and overcome the limitations of the automatic calibration procedure highlighted in the discussion section."

have been modified as (see Lines 614-618 of the revised manuscript):

"The application of the STREAM v1.3 model on a larger number of basins with different climatic- physiographic characteristics (e.g., including more arid basins, snow-dominated, lots of topography, heavily managed) would permit to investigate the possibility to regionalize the model parameters and overcome the limitations of the automatic calibration procedure highlighted in the discussion section."

TE2: It does not make sense to me why the inclusion of a tributary would actually cause the model to overestimate discharge. Rather, it seems that it would cause a discharge underestimate (but not including some additional inflow).

AC: I think the TE is referring to the overestimation of river discharge over section 7 and to the author reply to comment n.3 of RC2.

In the reply to RC2, the authors would underline that in section 7 there is not an overestimation of river discharge rather than an uncorrected river discharge simulation.

The reason of this uncorrected river discharge simulation over this section is likely due to the fact that the STREAM v1.3 model is calibrated with respect to the river discharge observed in section 6 (see Figure below or Figure 3 in the manuscript). Section 7 is located at the closure section of basin 41, a small portion of the entire basin closed to section 6 (green-colored basin). In this case, the characteristics of the basin 41 could be different from the ones of the entire basin closed to section 6 and this could be the cause of the incorrect river discharge estimation.

We hope that this reply has resolved the TE's doubts.

This aspect has been better specified in the revised manuscript (see Lines 477-482 of the revised manuscript):

"Over section 7, located over the Rock river, a relatively small tributary of Mississippi river (see Table 1), the STREAM v1.3 model overestimation has to be attributed to: 1) the different characteristics of the Rock river basin with respect to the entire basin closed to section 6 where the model has been calibrated (see Figure 3); 2) the small size of the Rock river basin (23'000 $km^2$, if compared with GRACE resolution, 160'000 $km^2$) for which the model accuracy is expect to be lower."

[Figure]

*Figure: Mississippi sub-basin delineation. Red dots indicate the location of the discharge gauging stations; different colours identify different inner sections (and the related contributing sub-basins) used for the model calibration*

TE3: The dams on the Mississippi are run-of-the-river dams. They do not have significant storage, nor do they substantially affect the discharge. I suggest that you take this into account when discussing the errors.

AC: We are aware that most of the dams along the Mississippi are run-of-the-river and they do not substantially affect the discharge. However, the dams which we are referring to in section 1 and 2 are the Garrison and Gavins dams (see Table 1), with a maximum storage of $29'383\times10^9$ and $0.607\times10^9$ m$^3$, respectively. As an example, the figure below by Skalak et al. (2013) represents the river discharge time series recorded at Bismark (section 1 in the manuscript) from 1930 to 2010. After the Garrison dam construction, a strong impact on river discharge is evident.

[Figure]

*Figure: Hydrograph for the stream gage at Bismarck (USGS 06342500). The year the Garrison Dam was completed is indicated separating pre- and post-dam flows. There is an increase in baseflows and decrease in peakflows as a result of the dam (by Skalak, K. J., Benthem, A. J., Schenk, E. R., Hupp, C. R., Galloway, J. M., Nustad, R. A., & Wiche, G. J. 2013. Large dams and alluvial rivers in the Anthropocene: The impacts of the Garrison and Oahe Dams on the Upper Missouri River. Anthropocene, 2, 51-64.).*

This aspect has been specified in the manuscript, Lines 169-170 have been modified as follows (see Lines 170-176 of the revised manuscript):

The basin is also heavily regulated by the presence of large dams (Global Reservoir and Dam Database GRanD, Lehner et al., 2011) most of them located on the Missouri river. In particular, the river reach between Garrison and Gavins Point dams is the portion of the Missouri river where the large main-channel dams have the greatest impact on river discharge providing a substantial reduction in the annual peak floods, an increase on low flows and a reduction on the overall variability of intra-annual discharges (Alexander et al., 2012)."

Lines 194-196 have been modified as follows (see Lines 199-204 of the revised manuscript):

"As it can be noted, mean annual river discharge ranges from 141 to 17'500 m$^3$/s, and 3 out 11 sections are located downstream big dams (Lehner et al., 2011). In particular, Garrison (the fifth-largest earthen dam in the world), Gavins Point and Kanopolis dams located downstream section 1, 2 and 5 respectively (see Figure 3 and Table 1), are three large dams with a maximum storage of 29'383×10$^9$ m$^3$, 0.607×10$^9$ m$^3$, and 1.058×10$^9$ m$^3$ respectively.

An additional remark has been added to Lines 467-475 of the revised manuscript:

"In particular, for river sections 1 and 2 even if KGE reaches values equal to 0.35 and 0.40 (for the whole period), respectively, there is not a good agreement between observed and simulated river discharge and the R score is lower than 0.55 for both river sections. The worst performance is obtained over section 5, with negative KGE and low R (high RRSME). These results are certainly influenced by the presence of large dams located upstream to

these river sections (i.e., Garrison, Gavins Point and Kanopolis dams, see Table 1) which have a strong impact on discharge: the model, not having a specific module for modelling reservoirs, is not able to accurately reproduce the dynamics of river discharge over regulated river sections."

REFERENCE:

Alexander, J. S., Wilson, R. C., and Green, W. R.: A brief history and summary of the effects of river engineering and dams on the Mississippi River system and delta (p. 53), US Department of the Interior, US Geological Survey, https://doi.org/10.3133/cir1375, 2012.

TE4: Point 7 on Fig. 3 appears to be behind the semi-transparent basin polygon. It is also substantially north of Keokuk, IA. Some numbers on this figure are obscured. Please clean it up and fact-check it.

AC: Figure 3 has been improved according the TE suggestion.

[Figure]

TE5: This is a point that I raised initially: Your notes about snowfall indicate mountainous regions, and yet a significant fraction of the snowmelt comes from the northern lowland regions in the Mississippi River watershed. Could you look through your article and double check that you are including a proper description of the hydrology, rather than applying some a priori assumptions that may come out of experiences in Europe?

AC:According the TE suggestion, we modified the description of the Mississippi river basin hydrology as follows:

The river flow has a clear natural seasonality mainly controlled by spring snowmelt (coming from the Missouri and the Upper Mississippi, the eastern and the upper part of the basin,

respectively, Dyer 2008) and by heavy precipitation exceeding the soil moisture storage capacity (mostly occurring in the eastern and southern part of the basin, Berghuijs et al., 2016). The basin is also heavily regulated by the presence of large dams (Global Reservoir and Dam Database GRanD, Lehner et al., 2011) most of them located on the Missouri river. In particular, the river reach between Garrison and Gavins Point dams is the portion of the Missouri river where the large main-channel dams have the greatest impact on river discharge providing a substantial reduction in the annual peak floods, an increase on low flows and a reduction on the overall variability of intra-annual discharges (Alexander et al., 2012).

Anyway, we would like to outline that the basin description does not impact the model results (driven by observation of precipitation, temperature, soil moisture and terrestrial water storage anomalies)

REFERENCES:

Dyer, J.: Snow depth and streamflow relationships in large North American watersheds, J. Geophys. Res., 113, D18113, https://doi.org/10.1029/2008JD010031, 2008.

Alexander, J. S., Wilson, R. C., & Green, W. R. (2012). A brief history and summary of the effects of river engineering and dams on the Mississippi River system and delta (p. 53). US Department of the Interior, US Geological Survey.

TE6: As such, I would also like you to re-address RC3, Line 262: The grid size of your model IS appropriate to simulate much of the snow in the northern reaches of the Mississippi basin, and the upper Mississippi above Keokuk (IA) contains significant snow. (I wonder if this might relate to some of the observed discrepancy between model and data noted in RC2, Comment 3.)

AC: Thanks for this comment. In our reply to RC2, comment 3 we meant that the scale of the model is not suitable for comparison against in situ point-scale snow observations (unless they would cover the 25 km pixel with hundreds of them which is not the case here)

However, in the manuscript we could evaluate the impact of snow on runoff. To do so and to in depth analyze the observed discrepancy between model and data noted in RC2 comment 3, we should select a snow-dominated basin and run the model with and without the snow module. Accordingly, we selected the basin closed to section 7 (basin 41 in Figure 3) and we calibrated the STREAM v1.3 model with and without the snow module. We obtained a good agreement against observed data (KGE=0.71) both by including or not the snow module as shown in the Figure below.

[Figure]

*Figure: Comparison between observed (green line) and simulated river discharge data at the outlet section 7, by including (red dashed line) or not (blue line) the snow module in the STREAM v1.3 model.*

This result indicates that:

1) the snow impact on runoff cannot be easily distinguished in the sections selected in the paper;

2) the uncorrected river discharge simulation over section 7, shown in the paper, is due to the "uncorrected" model parameters associated to basin 41. Likely, the characteristics of this basin are different from the ones of the entire basin closed to section 6 where the model has been calibrated (see reply to TE2).

This last point has been stressed in Lines 477-482 of the revised manuscript:

"Over section 7, located over the Rock river, a relatively small tributary of Mississippi river (see Table 1), the STREAM v1.3 model overestimation has to be attributed to: 1) the different characteristics of the Rock river basin with respect to the entire basin closed to section 6 where the model has been calibrated (see Figure 3); 2) the small size of the Rock river basin (23'000 km2 , if compared with GRACE resolution, 160'000 km2) for which the model accuracy is expect to be lower."

TE7: "Although the mascon size is smaller than the inherent spatial resolution of GRACE, the model exhibits a relatively high spatial resolution." It is highly resolved, yes, but I wonder if you are discussing accuracy more than resolution? Could you explain this a bit more?

AC: Sure. We would be happy to expand on this. Here, we are indeed discussing the spatial resolution of the model. A high spatial resolution of the model is attributed to an application of a Wiener filter. This filter makes use of full covariance matrices of noise and signal in GRACE data. The filter ensures that a minimal smoothing is applied to GRACE data. To that end, it exploits noise and signal covariance matrices in the spatial domain. The filtering is performed in the spatial domain, too. It is done in line with these noise and signal covariance matrices. This means that the higher signal-to-noise ratio in a particular area, the less smoothing is applied and the vice versa. This way, the filter avoids an aggressive smoothing

when it is not necessary. This leads to a higher spatial resolution of the model. This is discussed in details in (Klees et. al 2008), where evidences of a higher spatial resolution as a result of an application of such a filter are provided, too. We have now briefly incorporated these remarks in the updated manuscript and lines 222-226 in the manuscript:

"This is attributed to a statistically optimal Wiener filtering, which uses signal and noise covariance matrices. The coloured (frequency-dependent) noise characteristic of KBR data was taken in to account when compiling the model, which has allowed for a reliable computation of these noise and signal covariance matrices."

have been reformulated into the following (see Lines 227-235 of the revised manuscript):

"This is attributed to a statistically optimal Wiener filtering, which uses signal and noise full covariance matrices. This allows the filter to fine tune the smoothing in line with the signal-to-noise ratio in different areas. That is, the less smoothing, the higher signal-to-noise ratio in a particular area and vice versa. This ensures that the filtering is minimal and aggressive smoothing is avoided when unnecessary. Further details of such a filter can be found in (Klees et. al 2008). Importantly, the coloured (frequency-dependent) noise characteristic of KBR data was taken in to account when compiling the GRACE model, which has allowed for a reliable computation of the aforementioned noise full covariance matrices."

REFERENCE:

Klees, R., Revtova, E.A. , Gunter, B.C. , Ditmar, P., Oudman, E., Winsemius H.C., and Savenije H.H.G.: The design of an optimal filter for monthly GRACE gravity models, Geoph. J. Intern., 175 (2): 417–432, https://doi.org/10.1111/j.1365-246X.2008.03922.x, 2008.

TE8: Regarding RC3, Line 520: Could you please dig a bit deeper to address this question about human activities and GRACE, including looking into published works on reservoirs and GRACE? I realize that there is a large scale gap (so the spatial resolution will be poor) but these reservoirs can be quite significant -- as can human activities that affect soil moisture and groundwater storage.

AC: We checked the literature and we found support to our to RC3, Line 520 reply. Indeed, Longuevergne et al. (2013) or Deggim et al. (2021) clearly explain that GRACE can "see" mass changes due to large human controlled reservoirs or natural lakes with strong (seasonal) variations and/or trends. Specifically, Longuevergne et al. (2013) wrote:

"Virtually all reservoirs are point masses at the spatial resolution of GRACE. For example, a large reservoir with a typical surface area of $\sim$ 1000 $km^2$ (Garrison reservoir in the Mississippi is 1500 $km^2$) is about two orders of magnitude less than that of the smallest basin ($\sim$ 200 000 $km^2$) that can be typically resolved by GRACE observations. The precision of GRACE observations allows detection of 1 cm TWS change within a 200'000 $km^2$ basin (= 2 $km^3$ TWS change). This is comparable in mass (hence detectability) to a 2 m water level change within a 1000 $km^2$ reservoir".

However, they specify that "it has not been clear how small scale and/or discontinuous distributions of water sources affect basin-scale average water storage changes typically estimated from GRACE data". The main problem is that the mass changes do not necessarily appear exactly at the location of their origin and with the correct magnitude. Thus, they can distort the water storage estimate for neighbouring areas or the average over a river basin (Deggim et al., 2021).

For that, the use of GRACE data to take into account the human activity or the water extraction practices over a basin is not straightforward. A possible solution to address this problem is try to disentangle the reservoir/lakes impact on GRACE or use the RECOG RL01 product ad hoc developed by Deggim et al. (2021). However, this aspect is beyond the paper purpose.

REFERENCES:

Longuevergne, L., Wilson, C. R., Scanlon, B. R., & Crétaux, J. F. (2012). GRACE water storage estimates for the Middle East and other regions with significant reservoir and lake storage. Hydrol. Earth Syst. Sci. Discuss, 9(10), 11-131.

Deggim, S., Eicker, A., Schawohl, L., Gerdener, H., Schulze, K., Engels, O., ... & Longuevergne, L. (2021). RECOG RL01: Correcting GRACE total water storage estimates for global lakes/reservoirs and earthquakes. Earth Syst. Sci. Data, 13, 2227–2244. https://doi.org/10.5194/essd-13-2227-2021.

---

## Author Response (AR3)

**Review of "Synergy between satellite observations of soil moisture and water storage anomalies for global runoff estimation"**

We thank the Topical Editor (TE) and the two anonymous reviewers for their supportive review. In the following, the author replies (AC) to the TE and reviewers comments (red lines) are reported.
Italic text on the AC replies reports the changes made on the revised manuscript. Lines of the revised manuscript refer to the author's track-changes version of the manuscript.

**TEC: Comments and replies**

Dear Stefania Camici and co-authors,

based on the referee recommendations, I would like to see the following:

(1) A discussion of the data--model fit and considerations of how to improve this (re: Referee #2)

(2) A comparison and/or discussion of how STREAM compares to other similarly scoped and well-established models (I'll add PRMS and/or GSFLOW to the list, too) in order to identify why a modeler may want to use STREAM over these.

I am happy to discuss this further as you may find it helpful.

AC: We thank the TE for the helpfulness demonstrated with the authors. We modified the text to address the TE and reviewer comment's as specified in the following replies. Specifically, in the revised manuscript:

1) we added references to underline that the performances of the STREAM v1.3 model over the west part of the Mississippi river basin and in particular over the Great Plains are similar to the ones obtained by other hydrological models. Lines 475-477 have been modified as follows:

*In particular, over section 3 the STREAM v1.3 model overestimates the observed river discharge due the presence of large dams along the Missouri river, over the Great Plains region. This area is well known from other global hydrological models (e. g., ParFlow-CONUS and WRF-Hydro) to be characterized by low performances in terms of river discharge modelling (O'Neill et al., 2020, Tijerina et al., 2021).*

*O'Neill, M. M., Tijerina, D. T., Condon, L. E., and Maxwell, R. M., Assessment of the ParFlow–CLM CONUS 1.0 integrated hydrologic model: evaluation of hyper-resolution water balance components across the contiguous United States, Geosci. Model Dev., 14, 7223–7254, https://doi.org/10.5194/gmd-14-7223-2021, 2021.*

*Tijerina, D., Condon, L., FitzGerald, K., Dugger, A., O'Neill, M. M., Sampson, K., ... and Maxwell, R., Continental Hydrologic Intercomparison Project, Phase 1: A Large-Scale Hydrologic Model Comparison Over the Continental United States, Water Resour. Res., 57(7), e2020WR028931, https://doi.org/10.1029/2020WR028931, 2021.*

2) we better specified the strengths, the limitations, and the innovative aspects of the STREAM v1.3 model. For that Lines 531 to 589 have been modified as follows:

[revised manuscript text omitted]

**Referee Report #1: Comments and replies**

Line 162: replace "norther" with "northern"

AC: Accordingly, the text has been changed.

Line 235: replace "taken in to account" with "taken into account"

AC: Accordingly, the text has been changed.

**Referee Report #2: Comments and replies**

In this study, Camici and coauthors present a simplified conceptual discharge model that uses precipitation, soil moisture, and temperature to model quick runoff and GRACE-derived storage changes to model slow runoff. Although the rational behind the work is solid, though not particularly novel (https://www.sciencedirect.com/science/article/pii/S0012821X08006766, https://link.springer.com/chapter/10.1007/978-3-030-02197-9_1), the results are only good in the basin it's calibrated over, with little potential for transfer. The coauthors attempt to validate their model's utility by expressing it's easy of use, computational efficiency, and limited input data requirements, but it is far from the only model to check these boxes. Without comparison to some more commonly used models, say VIC, SWAT, Sacramento, or HEC-HMS, it's hard to convince people that they should use the presented STREAM model. I strongly encourage the coauthors to compare their results with other simplified conceptual discharge models to validate their model's utility.

AC: We modified the text to address the reviewer comment. Lines 531 to 589 have been modified as follows:

[revised manuscript text omitted]

R2: Line 278: The rain/snow differentiation model should be expanded on within the study. Rain/snow differentiation based on temperature and elevation is passably good, but at a large grid size like 25 x 25 km, the topographic complexity of higher elevations is lost. A differentiation scheme like that used in IMERG may be preferred, but isn't necessary. Still, this should be acknowledged, however briefly.

AC: We thank the reviewer to outline this aspect. In the manuscript we mentioned that (see Lines 285-290):

*"In particular, according to Cislaghi et al. (2020), SWE is modelled by using as input $T_{air}$ and a degree-day coefficient, $C_m$ , to be estimated by calibration. We have to acknowledge that, even though this rain/snow differentiation method works quite efficiently at a large grid size like the one used in the study (25 x 25 km), the topographic complexity of higher elevations can be lost. A different differentiation scheme based e.g., on the wet bulb temperature like in IMERG (Wang et al., 2019; Arabzadeh and Behrangi, 2021), would be preferable but is out of the purpose study."*

*References have been added to the revised manuscript.*

Wang, Y. H., Broxton, P., Fang, Y., Behrangi, A., Barlage, M., Zeng, X., and Niu, G. Y.: A wet-bulb temperature-based rain-snow partitioning scheme improves snowpack prediction over the drier western United States, Geophys. Res. Lett., 46(23), 13825-13835, https://doi.org/10.1029/2019GL085722, 2019.

Arabzadeh, A., and Behrangi, A.: Investigating Various Products of IMERG for Precipitation Retrieval Over Surfaces With and Without Snow and Ice Cover, Remote Sens., 13(14), 2726; https://doi.org/10.3390/rs13142726, 2021.

R2: Lines 345-348: Using a calibration tool would be preferable to manually adjusting to maximize Kling-Gupta. Perhaps one was used, but it is not specified. Also, does paragraph 5.1 relate to calibration, or is it paragraph 5.4?

AC: For the maximization of the Kling-Gupta efficiency Index we used a standard gradient-based automatic optimization method. This has been specified in the manuscript (see Lines 351-352)

*"For model calibration, a standard gradient-based automatic optimization method (Bober 2013) was used."*

The reference has been added tot the revised manuscript.

Bober, W. Introduction to Numerical and Analytical Methods with MATLAB for Engineers and Scientists; CRC Press, Inc.: Boca Raton, FL, USA, https://doi.org/10.1201/b16030, 2013.

R2: Section 5.1: "1. Input data collection" is unnecessary to include.

AC: Accordingly this part has been removed from the revised manuscript. In the new manuscript it can be read (see Lines 356-361):

*1. Sub-basin delineation. STREAM v1.3 model is run in the semi-distributed version over the Mississippi River basin. The TopoToolbox (https://topotoolbox.wordpress.com/), a tool developed in Matlab by Schwanghart et al. (2010), and the SHuttle Elevation Derivatives at multiple Scales (HydroSHED, https://www.hydrosheds.org/) DEM of the basin at the $3''$ resolution (nearly 90 m at the equator) have been used to derive flow directions, to extract the stream network and to delineate the drainage basins over the Mississippi River basin. In particular, by considering only rivers with order greater than 3 (according to the Horton-Strahler rules, Horton, 1945; Strahler, 1952), the Mississippi watershed has been divided into 53 sub-basins as illustrated in Figure 3.*

R2: Line 414-415: It is not clear to me what "to get to the right answers for the right reasons" means in this context and its tedious to hunt it down in the cited paper.

AC: This aspect has been expanded in the revised manuscript. The rationale behind the well-known concept "to get to the right answers for the right reasons" is that the hydrological models are today highly performing and able to reproduce a lot of hydrological variables. For that, the model performances should not only be evaluated against observed streamflow or associated signature measures, but complementary datasets representing internal hydrologic states and fluxes, such as soil moisture and evapotranspiration could be used to evaluate the capability of the model to simulate spatially distributed land surface fluxes controlled by local soil moisture availability and land surface hydrology.

Lines 414-415 have been modified as (see Lines 411-416 of the revised manuscript):

*"3.     External validation aimed to test the capability of the model "to get the right answers for the right reasons" (Kirchner 2006). The rationale behind this concept is that the hydrological models are today highly performing and able to reproduce a lot of hydrological variables. For that, the model performances should not only be evaluated against observed streamflow, but complementary datasets representing internal hydrologic states and fluxes, e.g., soil moisture, evapotranspiration, runoff etc) should be considered."*

R2: Line 500-501: I would encourage you to include a precipitation map as a figure to illustrate your point.

AC: We thank the reviewer for the suggestion. A figure, showing the mean annual precipitation data obtained by TMPA 3B42 V7 and GSWP3 datasets over the Mississippi river basin has been added to the supplementary material. As it can be noted, both the datasets identify a strong difference between the western (dry) and the eastern (wet) area of the basin.

[Figure]

*Figure. Mean annual precipitation data over the period 2003-2014 obtained by TMPA 3B42 V7 and GSWP3 datasets over the Mississippi river basin.*

R2: Line 595: By the author's own admission (Lines 486-490), the model may not be suitable to reproduce discharge in basins not calibrated over. This should be changed to something less absolute. "Under some circumstances, the STREAM model can be used to estimate discharge in basins not calibrated over, especially those without upstream dams with comparable size and land cover." Or something similar.

AC: We thank the reviewer for this helpful suggestion. The sentence has been modified as:

*"Conversely, the performances over river section 8, whose parameters have been set equal to the ones of river section 10, are quite high (KGE equal to 0.71, 0.80 and 0.77 for the entire, the calibration and the validation period, respectively; R equal to 0.83, 0.84 and 0.84 for the entire, calibration and validation periods, respectively). This outcome demonstrates that under some circumstances, the STREAM v1.3 model can be used to estimate river discharge in basins not calibrated over, especially those without upstream dams and with comparable size and land cover.*

*Although it is expected that the performances of STREAM v1.3 model, as any hydrological model calibrated against observed data, can decrease over the gauging sections not used for the calibration, the findings obtained above raises doubts about the robustness of model parameters and whether it is actually possible to transfer model parameters from one river section to another with different interbasin characteristics."*

---

## Author Response (AR4)

**Topical Editor**

Dear Dr. Camici and co-authors,

Thank you for your patience in my work to provide a response. I have been having a bit of difficulty in deciding what to do, considering the long time that your paper has been going back and forth through the editorial process alongside the fact that the new and independent referee feels, as do I, that the presentation of the work remains difficult to follow.

I was hoping to suggest more minor revisions, but based on the points of the new referee, added to my own concerns, I am suggesting "major revisions". Indeed, the new referee indicates similar questions and concerns to those that I have been having throughout the entire review process. If you respond to these fully and with appropriate changes to your paper, I would hope to be able to move forward more quickly with it. These include substantial improvements in the writing and consistency of your paper, without which I feel that others will find it difficult to understand (thereby significantly decreasing its impact), and appropriate contextualization of your work.

I am sorry that this news is not better: I realize that we have been at this for a while, and have been considering the best way to address these comments. Please feel free to be in contact if any additional clarification be needed,

R: We tried to do our best to make the manuscript and the revision more complete as possible. We hope we will be able to dispel all doubts of the reviewers and Topical Editor and to see the manuscript finally published on the GMD journal.

Specifically, we carried out the following additional analysis and modifications in the revision or in the revised manuscript:

- 1) A comparison between the modelled STREAM and PRMS river discharge data over the gauging stations selected in the Mississippi river basin. The analysis confirmed a good capability of the STREAM model to provide accurate river discharge estimates even better than those obtained through a comprehensive hydrological model such as the PRMS model (see reply to reviewer 1, below). However, as this analysis is beyond the scope of the manuscript, the authors prefer do not include it in the revised manuscript;
- 2) A sensitivity analysis of STREAM parameters. According to the suggestion of the Reviewer 2, a sensitivity analysis has been added to the revised manuscript in the paragraphs 5.4 and 6.3. This analysis allowed us to identify the most sensitive parameters of STREAM model and it poses the basis for the regionalization of model parameters;
- 3) According to the suggestions of reviewer 2, the STREAM model description and the related figure (Figure 2 in the revised manuscript) were accurately modified to better illustrate the model structure and the simulated processes;
- 4) The innovative aspect of the STREAM model has been in-deep highlighted. Through a comparison with conceptual hydrological models available in literature (see Lines 649-656) we

stressed the parsimonious structure of the model and the key role of satellite observations that allowed us to simplify the model structure (see Lines 625-636).

**Reviewer #1**

Dear Dr. Camici and co-authors,

Thank you for your revisions. However, you have chosen to only respond to select portions of the comments, which gives me no choice but to require further revisions. Furthermore, I have encountered some small errors and some unclear and/or misleading language in your revisions, which I have tried to correct, but I would ask that you take some care (or perhaps just place a bit of context in front of your enthusiasm at what you have built) in order to accurately describe the model's capabilities and limitations.

As a result, I have undertaken a read of your paper by myself. I am not fully familiar with all of the details involved in developing a hydrological model. For this reason, and because of the prior reviews (major revisions and reject, though I think that we agree that the latter's suggestion lies in contrast to their middling evaluation and lack of detail), I am going to send it out to one more referee. If the referee's decision is favorable, I will ask you to respond to both their reviews and mine.

Unanswered portions of my two requests:

(1) "considerations of how to improve this [the data--model fit]"

R: To answer this question in the previous revision of the manuscript we highlighted the possibility to regionalize the model parameters (see Lines 537-543; 684-698), avoiding to add any analysis related to the regionalization. The main reason behind this choice lies in the fact that the authors in the present manuscript would like to introduce the STREAM model and stress how satellite data can provide precious information in modeling river discharge leaving any additional analysis to future manuscripts. However, as this aspect seems crucial for the reviewer and the Topical Editor, we carried out a preliminary analysis to regionalize the model parameters.

For the application of the STREAM model in the Mississippi basin, we identified 53 sub-catchments each one with own characteristics in terms of land-cover, topography, climate, etc. Through the calibration procedure, for the same sub-catchments we identified 5 groups of parameters (each one for each calibration site). By using this parameter sets we tried a first attempt of parameters regionalization. In particular, based on the results of the sensitivity analysis (see paragraph 6.3 in the revised manuscript), we fixed some model parameters (the less sensitive, i.e., gamma, T, Cm, C and D) to constant values (equal to the mean values obtained for the 53 sub-catchments) while we looked for plausible relationships with the basin characteristics for the others (i.e., alpha, beta and m). Specifically, we linked alpha to the aridity index, beta and m to the bulk density.

The results of this first attempt of model parameter regionalization are shown in the following. Figure1R shows the performances, in terms of KGE, obtained through the calibrated parameter set (on the left upper map) and the variation on model performances obtained by considering the regionalized model parameters (on the right upper map). The bottom plot of Figure 1R shows the

modelled river discharge obtained by considering the calibrated (STREAM cal.) and the regionalized (STREAM reg.) parameter sets. Figure 2R illustrates the performances of the model over the sections used for the model calibration.

As expected, both the figures highlight a worsening of model results when the regionalized set of parameters are used for running the model. However, the deterioration is lower than 30% and it can be retained acceptable.

Figure 1R. Performances obtained through the STREAM calibrated/ regionalized (left/right map) parameter sets.